# Combined multidimensional single-cell protein and RNA profiling dissects the cellular and functional heterogeneity of thymic epithelial cells

Fabian Klein [1], Clara Veiga-Villauriz [1], Anastasiya Börsch[2], Stefano Maio [1], Sam Palmer[3], Fatima Dhalla[1], Adam E. Handel[1,4], Saulius Zuklys[5], Irene Calvo-Asensio [5], Lucas Musette[5], Mary E. Deadman[1], Andrea J. White[6], Beth Lucas [6], Graham Anderson [6] & Georg A. Holländer [1,5,7] ✉

The network of thymic stromal cells provides essential niches with unique molecular cues controlling T cell development and selection. Recent single-cell RNA sequencing studies have uncovered previously unappreciated transcriptional heterogeneity among thymic epithelial cells (TEC). However, there are only very few cell markers that allow a comparable phenotypic identification of TEC. Here, using massively parallel flow cytometry and machine learning, we deconvoluted known TEC phenotypes into novel subpopulations. Using CITEseq, these phenotypes were related to corresponding TEC subtypes defined by the cells' RNA profiles. This approach allowed the phenotypic identification of perinatal cTEC and their physical localisation within the cortical stromal scaffold. In addition, we demonstrate the dynamic change in the frequency of perinatal cTEC in response to developing thymocytes and reveal their exceptional efficiency in positive selection. Collectively, our study identifies markers that allow for an unprecedented dissection of the thymus stromal complexity, as well as physical isolation of TEC populations and assignment of specific functions to individual TEC subtypes.

The thymus is essential for the formation and maintenance of the adaptive immune system. Its stroma provides a unique microenvironment promoting the generation and selection of T lymphocytes tolerant to an individual's own tissue antigens yet responsive to an unlimited range of pathogens or malignantly transformed cells. Thymic epithelial cells (TEC) constitute the major cellular element of the stromal scaffold[1–3]. Other cellular components of the stroma are different mesenchymal cell types and endothelial cells[2,4]. TEC attract blood-borne lymphoid progenitors, commit them to a T cell fate, provide the molecular cues essential for expansion and differentiation, and shape the T cell antigen receptor (TCR) repertoire via stringent processes of positive and negative selection based on the cells' antigen specificity[5–7].

The TEC compartment is composed of separate cortical (c) and medullary (m) lineages which have typically been defined by the cells' anatomical location, a limited number of phenotypic markers

[1]Department of Paediatrics and Institute of Developmental and Regenerative Medicine, University of Oxford, Oxford, UK. [2]Department of Biomedicine, University of Basel, Basel, Switzerland. [3]Mathematical Institute, University of Oxford, Oxford, UK. [4]Nuffield Department of Clinical Neurosciences, University of Oxford, Oxford, UK. [5]Paediatric Immunology, Department of Biomedicine, University of Basel and University Children's Hospital Basel, Basel, Switzerland. [6]Institute for Immunology and Immunotherapy, Medical School, University of Birmingham, Birmingham, UK. [7]Department of Biosystems Science and Engineering, ETH Zurich, Basel, Switzerland. ✉e-mail: georg.hollander@paediatrics.ox.ac.uk

and several functional characteristics[8–10]. The surface marker Ly51 and reactivity to UEA1 are used to distinguish between cTEC (Ly51+UEA1−) and mTEC (Ly51−UEA1+). Markers such as CD80 and MHCII identify subsets of mTEC such as immature (CD80lo MHCIIlo; mTEClo) and mature epithelia (CD80hi MHCIIhi; mTEChi). The latter cells are further differentiated based on the cells' capacity to express the Autoimmune Regulator (Aire)[11,12]. Recent single-cell RNA-sequencing (scRNAseq) uncovered a remarkable TEC heterogeneity which could previously not be appreciated using the few cell surface markers available for flow cytometry[13–15]. For instance, a scRNAseq analysis of the TEC compartment of 4-week-old mice demonstrated a single cortical TEC type but 4 separate mTEC sub-types, namely immature and a mature mTEC, post-Aire mTEC and tuft-like mTEC[15]. Investigations of TEC heterogeneity across the life

trajectory (1–52-weeks of age) identified 9 different TEC subtypes whose relative frequencies vary with age[13]. For example, the cTEC compartment is composed of at least two main subtypes, desig-nated perinatal and mature cTEC[13]. Perinatal cTEC represent a major subpopulation early in life (~40% of all TEC in the first week after birth) but their relative frequency rapidly decreases thereafter with only a small fraction of these cells being detected in adult animals. Conversely, mature cTEC increase in frequency and represent the majority of cortical epithelia from 4 weeks of life onwards[13]. Inter-typical TEC are characterised by a gene expression profile that includes signatures typical for both cortical and medullary thymic epithelial lineages. They also express genes including *Pdpn*, *Ccl21a*, *Ly6a*, and *Plet1* that have previously been associated with mTEC, thought to have a progenitor potential, and localised at the

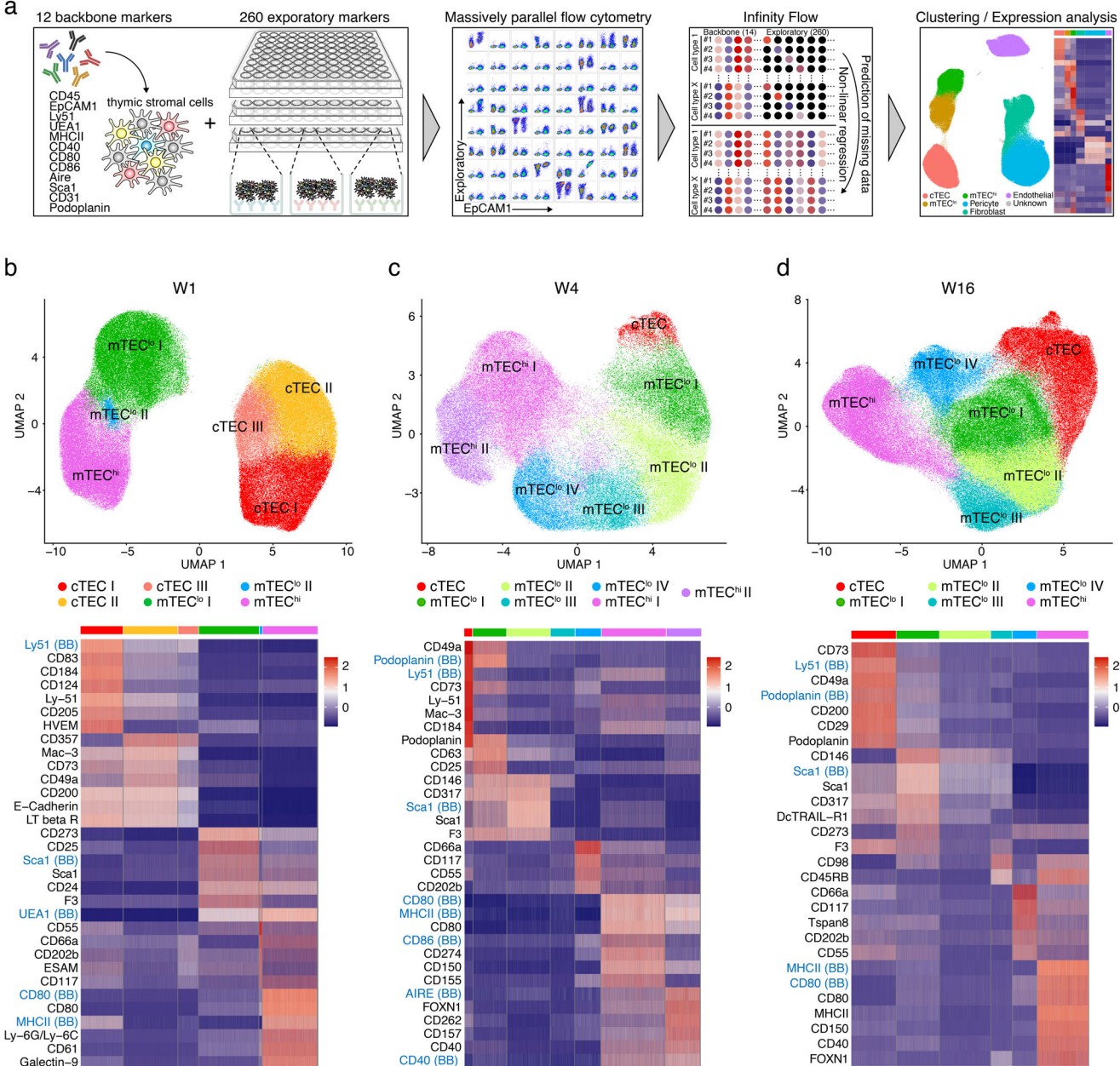

**Fig. 1 | Infinity Flow analysis reveals TEC heterogeneity. a** Schematic illustration of the surface marker screening pipeline. **b**–**d** Infinity Flow analysis was used to impute the expression of surface markers on TEC (CD45−EpCAM1+) derived from thymi of (**b**) 1- (*n* = 23), (**c**) 4- (*n* = 7), and (**d**) 16-week-old (*n* = 12) mice. Hierarchical clustering analysis was performed on (**b**) 182123, (**c**) 92402, and (**d**) 183124 TEC, respectively, and projected in a two-dimensional space using UMAP (top panels; 6–7 clusters were obtained per timepoint). Each colour represents a specific cluster as indicated. Heatmaps (bottom panels) display the expression of the top 7 markers upregulated in each cluster (log fold-change > 0.2). Backbone (BB) markers have a blue font.

cortico-medullary junction (CMJ)[16–20]. However, only few of these transcriptionally defined TEC subsets can currently be assigned to any cytometrically characterised TEC subpopulation, (for clarity, we refer to transcriptionally defined TEC clusters as subtypes and cytometrically specified TEC as subpopulations). Moreover, the hitherto absence of suitable and informative cell surface markers to physically isolate most of the TEC subtypes for in vitro analyses and in vivo transfer studies has disallowed to date further functional characterizations of these cells and the physical establishment of direct precursor-progeny relationships at single-cell resolution.

To address this limitation, we sought to screen mouse TEC for the expression of 260 cell surface markers employing massively parallel flow cytometry and the Infinity Flow computational pipeline to infer a co-expression pattern for any of the tested epitopes[21]. This approach identified several novel TEC surface markers that when suitably combined identified perinatal cTEC, intertypical TEC and tuft-like mTEC which had previously only been classified either by their distinct RNA expression profiles or a combination of cell surface and intracellular markers[22,23]. The identity of these phenotypically defined TEC subpopulations was verified by simultaneous measurements of mRNA and surface protein expression using CITEseq and, in the case of perinatal cTEC, further characterised functionally, spatially, and developmentally.

## Results

### Establishment of a cell surface expression atlas across thymic stromal cell subsets

To resolve thymic stroma heterogeneity at a phenotypic level, we sought to identify new cell surface markers that reliably and accurately identify TEC subsets hitherto only defined by the cells' individual gene expression profiles. For this purpose, we used massively parallel flow cytometry for 260 individual cell surface markers followed by an analysis employing Infinity Flow, a computational machine learning algorithm[21]. Thymic stomal cells were isolated as single cells from 1-, 4-, and 16-week-old mice, physically enriched and subsequently stained for 12 backbone markers that either alone or in combination reliably identified haematopoietic (CD45), different epithelial (EpCAM1, Ly51, UEA1, MHCII, CD40, CD80, CD86, Sca1, AIRE, Podoplanin), endothelial (CD31) and some mesenchymal cells (Sca1, Ly51, Podoplanin; Fig. 1a). As a next step, cells were split into aliquots and stained individually for 260 exploratory markers (Supplementary Table 1). Following sample acquisition, Infinity Flow was used to impute the expression level of each of the exploratory markers tested at single-cell resolution[21]. The resultant predictions of expression are based on non-linear functions of the recorded backbone markers. The observed heterogeneity and co-expression patterns were further analysed and visualised by the single-cell analysis pipeline Seurat[24]. Hierarchical clustering of the data resulted in 7 clusters for data drawn from 1-week-old mice and in 10 clusters for that of older animals, as illustrated in two dimensions by a Uniform Manifold Approximation and Projection (UMAP) (Supplementary Fig. 1a–c).

At each of the three separate timepoints, the major thymic stromal cell types, epithelia, fibroblasts, pericytes, and endothelial cells, could reliably be identified based on the expression of key markers including EpCAM1 (CD326) identifying TEC, CD140a marking fibroblasts, Ly51 and CD146 singling out pericytes, and CD31 staining endothelial cells. Additional markers identified subpopulations within these cell clusters (Supplementary Fig. 1a–c). Several of the antibody specificities to detect backbone epitopes were also included among the selected 260 exploratory markers (e.g., EpCAM1, CD31, Ly51, and Sca1) which allowed direct comparisons between exploratory and identical backbone markers, thus verifying the utility of the Infinity Flow algorithm. For these markers, we noted highly similar expression profiles, therefore demonstrating the reliability of the computational approach taken (Supplementary Fig. 1d).

The initial expression analysis not only confirmed by flow cytometry the heterogeneity among thymic stromal cell types, but also revealed a dynamic change over time in the relative representation of individual TEC subpopulations (Supplementary Fig. 1a–c). In a second analysis, we focused exclusively on EpCAM1+ cells and disclosed in 1-week-old but not older mice three separate cTEC subclusters as defined by the cells' differential expression of Ly51 and UEA1 thus illustrating a greater heterogeneity of the cTEC population early in life (Fig. 1b–d and Supplementary Fig. 2a–d). In mice 4 weeks of age and older, mTEC with a low surface expression of MHCII (designated mTEC$^{lo}$) segregated into 4 separate subclusters based on the differential expression of the surface markers analysed (Fig. 1b–d).

### cTEC heterogeneity identified by differential cell surface marker expression

We next queried whether the expression of CD83, CD40, HVEM (CD270), and Ly51 unequivocally classified individual cTEC subpopulations, since their intensity profile differed across cTEC clusters identified in 1-week-old mice (Fig. 2a). The expression of CD40 and HVEM were exclusively restricted to a subcluster designated cTEC I (see below) whereas the two other markers were detected across all cTEC subclusters, but with a stronger signal on the cTEC I (Fig. 2a).

Analysing a previously published scRNAseq dataset of TEC[13], transcripts for *Cd83*, *Cd40*, and *Enpep* (encoding Ly51) were detected in perinatal cTEC, albeit at various levels (Fig. 2b, c). In contrast, transcripts for *Tnfrsf14*, the gene encoding HVEM, were detected in only a few TEC but across several clusters, thus failing to unequivocally identify perinatal cTEC. This finding highlighted the limitations of gene expression studies to identify surface markers that matched the cells' RNA profile. To further assess the relationship of the cTEC I subcluster to TEC subtypes identified by scRNAseq, we generated a score of similarity using SingleR which related the RNA expression profile of individual cells to the computed cell surface expression pattern of cTEC I. This analysis demonstrated the highest similarity score to the pairing of cTEC I with perinatal cTEC (Fig. 2d).

We then aimed to define surface markers that allow the isolation of perinatal cTEC by flow cytometry. We used the presence of markers highly expressed in cluster cTEC I of 1-week-old mice while excluding markers detected on the majority of mature cTEC isolated from 4- to 16-week-old animals (Figs. 1b–d; 2a), as the population of cortical epithelia in older animals only includes perinatal cTEC at a very low frequency. Within the population of cTEC, we identified a subpopulation of cells that concomitantly expressed CD83 and CD40 but were Sca1 negative early in postnatal life (Fig. 2e; Supplementary Fig. 3a). As early as 4 weeks postnatally, TEC with a Sca1+ phenotype appeared among CD83+CD40+ cTEC. These cells were electronically excluded from further analysis as they represent mature cTEC that accumulate with age. The frequency of CD83+CD40+Sca1− epithelia changed substantially during the life course as the cells' relative representation progressively increased throughout organogenesis, plateaued in 1-week-old mice ($4.6 \times 10^4 \pm 1.4 \times 10^4$ cells) and subsequently decreased, displaying the lowest representation in 8-week-old animals ($2.4 \times 10^3 \pm 2.3 \times 10^3$ cells; Fig. 2e, f; Supplementary Fig. 3b).

Perinatal cTEC displayed in contrast to other cTEC subpopulations, comprising cTEC clusters II and III, higher cell surface levels for HVEM, Ly51 and MHCII which allowed the identification of these cells in combination with a high cell surface expression of CD83 and CD40, in the absence of Sca1. Perinatal cTEC also showed a higher *Foxn1* promoter activity as demonstrated in reporter mice where the expression of GFP is under the transcriptional control of the *Foxn1* locus (Fig. 2g).

### Identification of intertypical and tuft-like TEC

The Infinity Flow analysis of adult mice revealed 4 distinct mTEC$^{lo}$ clusters. Clusters I and II were observed in 4- and 16-week-old mice and displayed a similar expression profile for most of the 260 exploratory

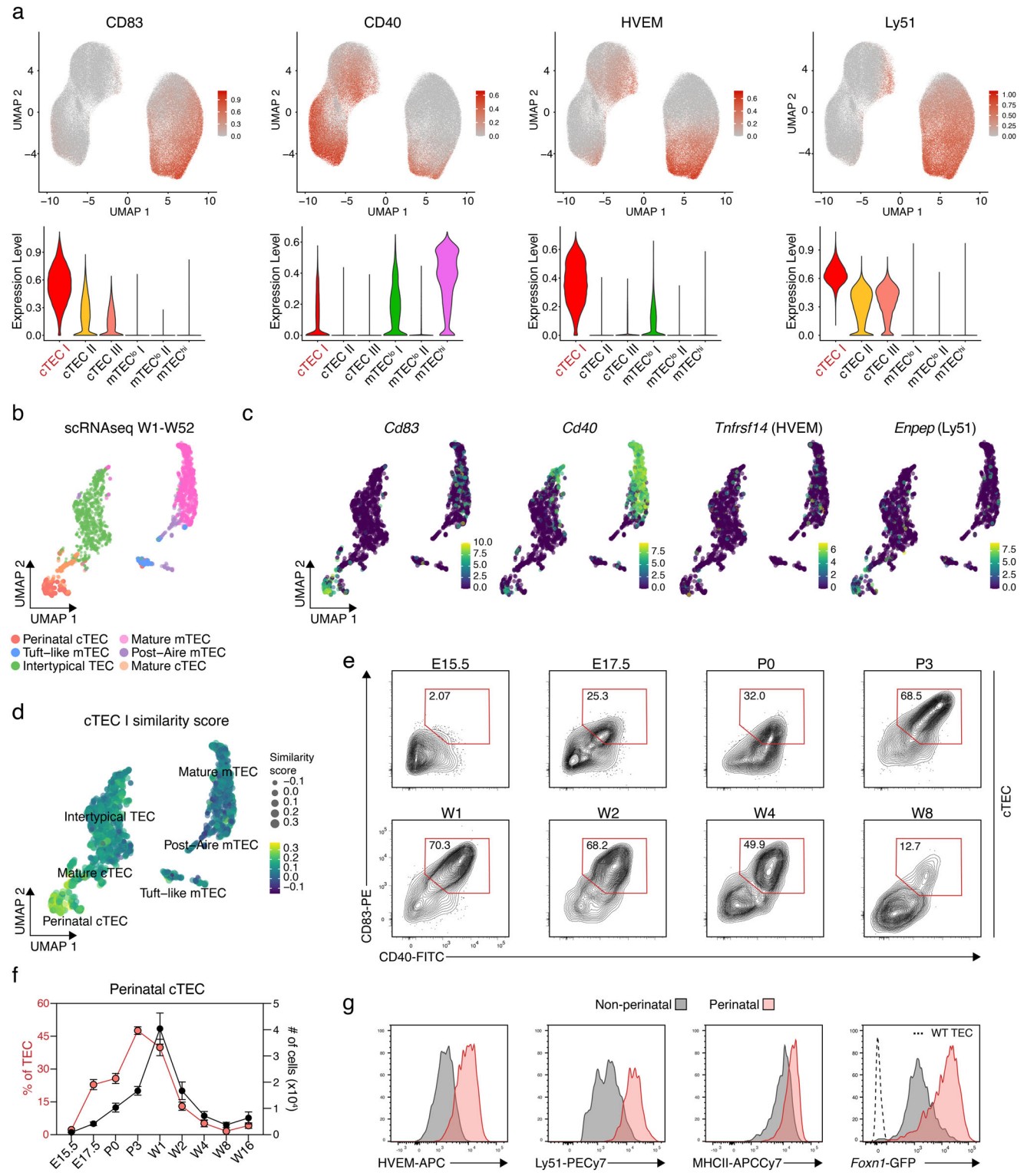

markers (Fig. 1b–d) including a shared expression of Sca1 and CD146 (Fig. 3a, b). To match these two phenotypically defined subpopulations to their corresponding transcriptome-determined TEC subtypes, we probed the RNA profiles of single TEC for the expression of *Ly6a/Ly6e* (encoding Sca1) and *Mcam* (encoding CD146). While transcripts for *Ly6a/Ly6e* were detected especially among intertypical TEC[13], *Mcam*-specific RNA was only detected at low levels and in different TEC, but mostly within the intertypical TEC subtype (Fig. 3c). This subtype is characterised by transcriptional features characteristic of both cTEC and mTEC –as phenotypically defined by the conventional surface

marker Ly51 and UEA-reactivity– contribute to this unique TEC subtype[13]. Using again the SingleR package, the similarity scores for both 4- and 16-week-old mTEC^lo I and II were calculated to be the highest when matched to the intertypical TEC subtype (Fig. 3d; Supplementary Fig. 4a). The mTEC^lo subpopulation contained cells that co-expressed Sca1 and CD146 and cells with this phenotype increased with postnatal age (Fig. 3e, f; Supplementary Fig. 4b). Although initially only detected among mTEC^lo this subpopulation was increasingly also observed within the cTEC compartment of mice older than 4 weeks of age (Fig. 3e, f).

**Fig. 2 | Surface expression profile of perinatal cTEC. a** UMAP graphs (top panels) and violin plots (bottom panels) illustrating the expression of CD83, CD40, HVEM, and Ly51 on TEC from 1-week-old mice. Colour gradient indicates expression levels in the UMAP graphs and colours in the violin plots represent the different clusters, as defined in Fig. 1b. **b** Hierarchical clustering analysis was performed on scRNAseq data obtained from TEC derived from 1-, 4-, 16-, 32, and 52-week-old mice and projected in a two-dimensional space using UMAP. **c** UMAP graphs illustrating the scaled expression of *Cd83*, *Cd40*, *Tnfrsf14* (HVEM), and *Enpep* (Ly51). Colour gradient indicates expression levels. **d** UMAP graph illustrating the similarity score of the cTEC I cluster from the 1-week Infinity Flow dataset to each cell of the scRNAseq reference dataset, based on the surface protein expression levels imputed by Infinity Flow. **e**, **f** Abundance of a CD83+CD40+Sca1− population (hereafter perinatal

cTEC) within cTEC (CD45−EpCAM1+UEA1−) was analysed at the indicated timepoints in WT C57BL/6 mice. Shown are (**e**) representative FACS plots of CD83 and CD40 expression and (**f**) cumulative data depicting the percent of perinatal cTEC within TEC as well as their total cell numbers (E15.5 $n = 13$, E17.5 $n = 7$, P0 $n = 6$, P3 $n = 8$, W1 $n = 7$, W2 $n = 7$ (percent of TEC) or $n = 6$ (number of cells), W4 $n = 5$, W8 $n = 5$, W16 $n = 8$, from 2 to 3 independent experiments per timepoint). Data are presented as mean values +/− SEM. Source data are provided as a Source Data file. E = embryonic day; P = postnatal day; W = postnatal week. **g** Representative histograms showing the expression of HVEM, Ly51, MHCII, and *Foxn1*-GFP within perinatal (CD83+CD40+Sca1−) and non-perinatal (CD83−CD40−) cTEC in 2-week-old C57BL/6 WT (HVEM, Ly51, MHCII, and *Foxn1*-GFP) and *Foxn1*GFP (*Foxn1*-GFP) mice.

A unique set of surface markers that specifically recognised the mTEC^lo cluster III could not be found. However, the simultaneous expression of CD66a and CD117 in the absence of Sca1 and CD63 positivity identified mTEC^lo cluster IV. This cluster was only detected in adult mice (Fig. 4a; Supplementary Fig. 5a), although the defining 4 cell surface markers could also be detected in the mTEC^lo cluster II of 1-week-old animals (Supplementary Fig. 5a). It is therefore possible that cluster mTEC^lo II of 1-week-old mice represents epithelia that form the separate cluster mTEC^lo IV in older animals. The single-cell transcriptomic analysis only partially matched the phenotypic analysis of mTEC^lo clusters as *Ly6a/Ly6e*- and *Cd63*-specific RNA could indeed be detected in the vast majority of TEC (Figs. 3c and 4b) whereas transcripts for *Ceacam1* (encoding CD66a) and *Kit* (encoding CD117) were largely absent in these cells (Fig. 4b). The similarity score revealed the best match between mTEC^lo cluster IV and post-AIRE and tuft-like mTEC (Fig. 4c; Supplementary Fig. 5b).

We next sought to define a phenotypic profile of cells belonging to cluster mTEC^lo IV that would allow their physical isolation by flow cytometry. For this purpose, we identified within the Sca1−CD63− mTEC^lo a subpopulation of cells that stained positively for both CD66a and CD117, thus mirroring the features identified for mTEC^lo cluster IV (Fig. 4d). Most of the Sca1−CD63−CD66a+CD117+ mTEC^lo cells (~70%) also expressed the serine/threonine-protein kinase Dclk1 (Fig. 4e) which was previously identified as a typical intracellular marker for tuft-like mTEC[15,23]. While staining for the tuft-like mTEC marker L1CAM was not successful in our hands, both the Dclk1 negative and positive fractions shared the CD104^lo tuft-like mTEC phenotype as previously described (Supplementary Fig. 5c)[15]. Furthermore, the presence of Dclk1 negative cells within tuft-like mTEC is supported by previous scRNAseq results[13,22]. We also observed that an absence of Dclk1 in Sca1−CD63−CD66a+CD117+ mTEC^lo correlated with a lower surface expression of CD66a and CD117, indicating that these cells are not yet fully differentiated into tuft-like mTEC (Supplementary Fig. 5d). Conversely, the vast majority of Dclk1-positive TEC were detected among Sca1−CD63−CD66a+CD117+ mTEC^lo (Supplementary Fig. 5e). In the absence of the transcription factor Pou2f3 Dclk1 expression was absent and Sca1−CD63−CD66a+CD117+ mTEC^lo cells were not detected (Fig. 4f, g) thus confirming their identity as tuft-like thymic epithelia[22,23]. Using these phenotypic features, we noted the presence of tuft-like mTEC to change over time with the highest frequency and cellularity in 4-week-old mice (Fig. 4h, i). As CD66a−CD117− cells ("non-tuft-like") made up a substantial portion (~40% at 4 weeks) of the relatively few Sca1−CD63− mTEC^lo, we sought to further determine their identity via bulk RNAseq. Specifically, we investigated whether these cells shared any transcriptomic features characteristic of tuft-like mTEC. This analysis revealed major transcriptional differences between the non-tuft-like and tuft-like cells. Genes associated with tuft-like cells were enriched in the CD66a+CD117+ fraction of cells and included *Dclk1*, *Il25*, *Ceacam1*, and *Kit* (Supplementary Fig. 5f). Similarly, the top genes identified in a previous scRNAseq experiment[13] defining tuft-like mTEC were expressed at a very high level in these tuft-like cells whilst their

transcripts were not present in non-tuft-like cells (Supplementary Fig. 5g). Correlation of the entire non-tuft-like transcriptome to the annotated TEC subsets confirmed this delineation from cells identified as tuft-like mTEC (Supplementary Fig. 5h). Because the CD66a−CD117− non-tuft-like mTEC displayed a gene expression profile that did not match any of the known mTEC subpopulations it is likely that these rare cells (~4% of total TEC) represent a mixture of mTEC subpopulations.

Tuft-like mTEC originate from mTEC that have once expressed the tissue restricted antigen Csnb[15]. We utilised *Csnb*^Cre::Rosa26^LSL-YFP reporter mice[15] that allow in vivo fate mapping within the mTEC lineage to test whether Sca1−CD63−CD66a+CD117+ mTEC^lo originate from a *Csnb*-expressing precursor. In keeping with the previous study, we identified 70–80% of the Sca1−CD63−CD66a+CD117+ mTEC^lo mTEC to be YFP labelled (Supplementary Fig. 5i), suggesting that they represent bona fide tuft-like mTEC. The mTEC^lo compartment is composed of medullary epithelia that either have not yet expressed the transcriptional facilitator AIRE or, alternatively, belong to a group of cells that have differentiated from AIRE-positive, mature mTEC. Tuft-like mTEC have previously been shown to derive, at least in part, from AIRE-positive precursors[22] and be enriched for expression of the surface glycoprotein Tspan8, an AIRE-enhanced tissue-restricted antigen[14,25]. We therefore tested whether Sca1−CD63−CD66a+CD117+ mTEC^lo (i.e. tuft-like) mTEC are positive for the expression of Tspan8. As many as 60% of tuft-like mTEC expressed Tspan8, further validating their identity and identifying these cells to contain post-AIRE mTEC (Supplementary Fig. 5j). The combined expression of YFP and Tspan8 was only detected in a small fraction of Sca1+CD146+ mTEC^lo intertypical TEC (Supplementary Fig. 5i,j), suggesting they belong to a TEC developmental stage that does not yet promiscuously express tissue-specific antigens.

## CITEseq validates novel TEC markers

We next concurrently determined the abundance of surface proteins and mRNA expression in individual thymic stromal cells using Cellular Indexing of Transcriptomes and Epitopes by sequencing (CITEseq)[26]. This approach was also taken to confirm unequivocally that the combined use of the newly described cell surface markers identifies specific and previously defined TEC subtypes. For this purpose, CD45−Ter119− cells were isolated from thymi of 1- and 16-week-old mice and stained with oligonucleotide-coupled antibodies each directed against EpCAM1, MHCII, UEA1, CD80, CD86, CD40, CD83, HVEM, CD73, Sca1, CD63, CD117, CD200, CD54, CD49a, CD274, CD9, Ly6G/Ly6C, Podoplanin, or CD31, respectively. Labelled cells were further subjected to single-cell sequencing along with estimating the abundance of antibody-derived tags (ADTs)[26].

Cell clustering resulted in the identification of 12 clusters with data computed either from single-cell gene expression profiling or, alternatively, from ADT, and displayed by means of t-distributed stochastic neighbour embedding (tSNE) (Supplementary Fig. 6a−c). The cell-type annotation performed was based on the gene expression

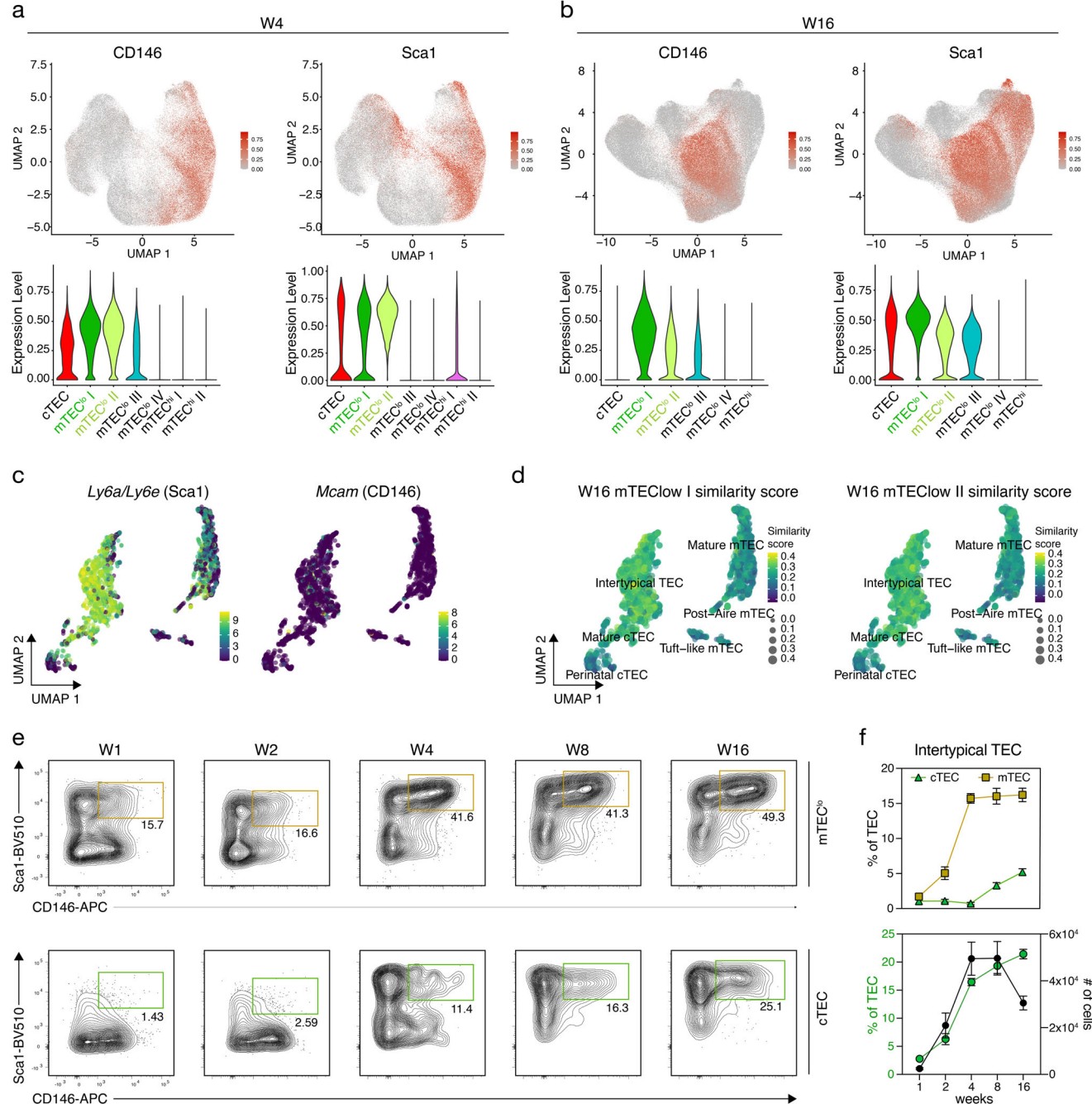

**Fig. 3 | Identification of intertypical TEC within cTEC and mTEC. a, b** UMAP graphs (top panels) and violin plots (bottom panels) illustrating the expression of CD146 and Sca1 on TEC from (**a**) 4- and (**b**) 16-week-old mice. Colour gradient indicates expression levels in the UMAP graphs and colours in the violin plots represent the different clusters, as defined in Fig. 1b. **c** UMAP graphs illustrating the scaled expression of *Ly6a/Ly6e* (Sca1) and *Mcam* (CD66a) in the scRNAseq dataset introduced in Fig. 2b. Colour gradient indicates expression levels. **d** UMAP graph illustrating the similarity score of the mTEC[lo] I and mTEC[lo] II clusters from the 16-week Infinity Flow dataset to each cell of the scRNAseq reference dataset, based on the surface protein expression levels imputed by Infinity Flow. **e, f** Abundance of a Sca1 and CD146 double positive population (hereafter intertypical TEC) within

mTEC[lo] (**e**; top panels) and within cTEC (**e**; bottom panels) was analysed at the indicated timepoints in C57BL/6 WT mice (for gating see Supplementary Fig. 4b). Shown are (**e**) representative FACS plots and (**f**) cumulative data for the percent of intertypical TEC within mTEC and cTEC (top panel) (W1 $n = 4$, W2 $n = 6$, W4 $n = 8$ (cTEC) or $n = 4$ (mTEC), W8 $n = 6$, W16 $n = 5$) and percent of intertypical TEC within TEC as well as their total cell numbers (bottom panel) (W1 $n = 4$, W2 $n = 6$ (percent of TEC) or 4 (number of cells), W4 $n = 4$, W8 $n = 6$, W16 $n = 5$, from 2 to 3 independent experiments per timepoint). Gating required adjustments between the different timepoints analysed due to age-dependent changes in surface expression levels of Sca1. Data are presented as mean +/− SEM. Source data are provided as a Source Data file.

profiles derived from the Immunological Genome Project (ImmGen)[27] and confirmed the identity of individual clusters as endothelial cells, fibroblasts, stromal cells, and epithelial cells, respectively. Hence, the chosen combination of selected surface markers was sufficient to identify individual stromal cell types (Supplementary Fig. 6d, e).

Analysing the captured gene expression profiles of only TEC identified 8 separate clusters (A-H), whereas examining the ADT data recognised 9 (1–9) clusters (Fig. 5a, b). Comparing the two approaches demonstrated a nearly pairwise relationship (Fig. 5c). The few exceptions observed concerned on one hand the clusters D and E which

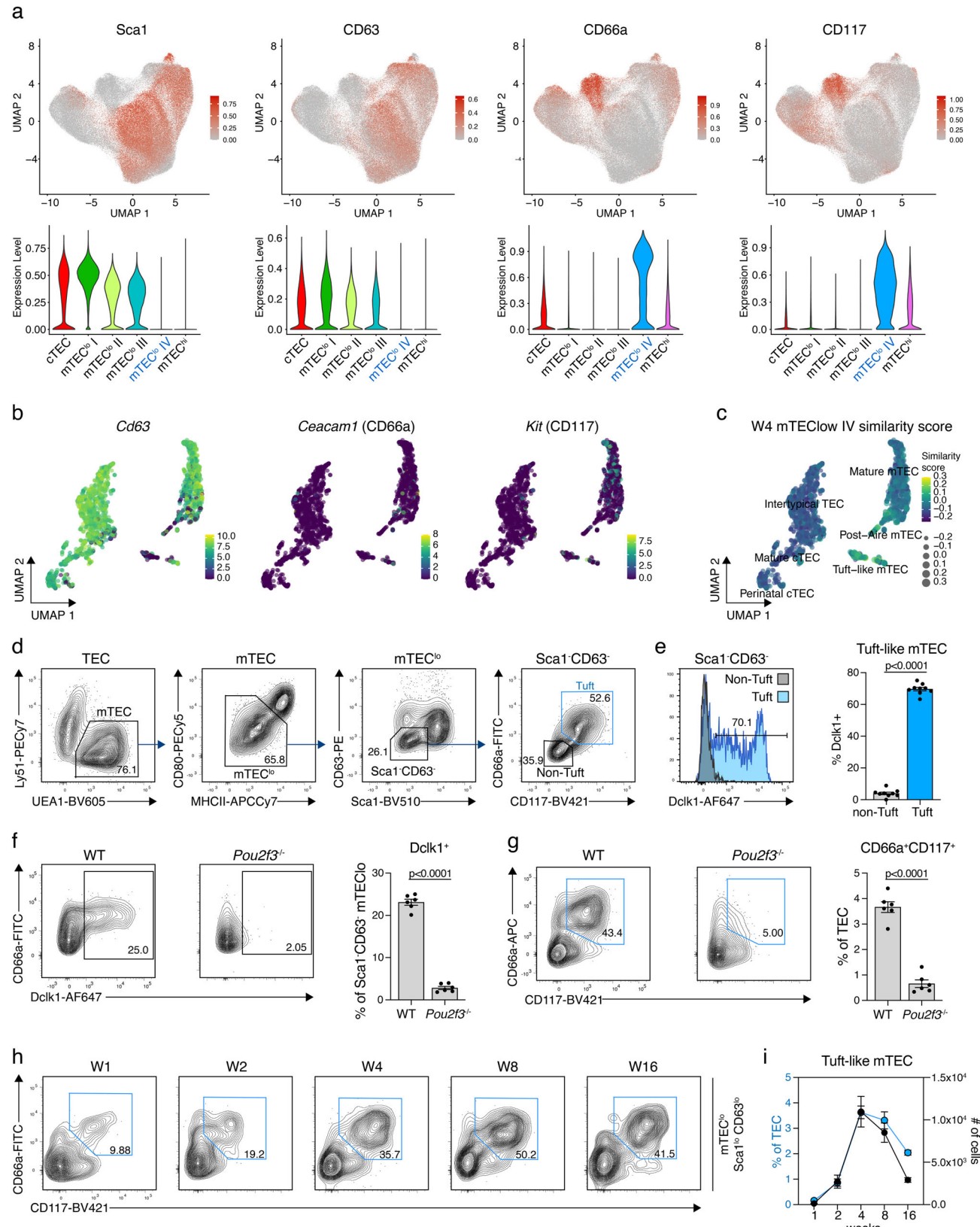

represented a mixture of clusters 4 and 5 and on the other hand cluster G which split into clusters 7 and 8 as a result of the clusters' differential expression of Ly6C/Ly6G (Fig. 5d). The limited number of antibodies used in the CITEseq analysis identified three cTEC cluster (defined as UEA1 non-reactive cells:1 corresponding to cluster A [1/A], 2/B, and 3/C), three mTEC$^{lo}$ (MHCII$^{lo}$CD80$^{lo}$: 4/D, 5/E, and 9/H) and three mTEC$^{hi}$

subpopulations (MHCII$^{hi}$CD80$^{hi}$: 6/F, 7/G, and 8) (Fig. 5d–f and Supplementary Fig. 6f).

We next assigned CITEseq-defined TEC cluster identities to those subtypes we have previously classified[13]. Clusters 1/A and, to a lesser extent, 2/B corresponded to perinatal cTEC. Cluster 2/B was further related to mature cTEC and cluster 3/C to mature cTEC and intertypical

**Fig. 4 | A combination of surface markers to define tuft-like mTEC. a** UMAP graphs (top) and violin plots (bottom) illustrating the expression of Sca1, CD63, CD66a, and CD117 on TEC from 16-week-old mice. Colour gradient indicates expression levels in the UMAP graphs and colours in the violin plots represent the different clusters, as defined in Fig. 1b. **b** UMAP graphs illustrating the scaled expression of *CD63*, *Ceacam* (CD66a), and *Kit* (CD117) in the scRNAseq dataset introduced in Fig. 2b. Colour gradient indicates expression levels. **c** UMAP graph illustrating the similarity score of the mTEC$^{lo}$ IV cluster from the 4-week Infinity Flow dataset to each cell of the scRNAseq reference dataset, based on the surface protein expression levels imputed by Infinity Flow. **d** Gating strategy to identify tuft-like mTEC within CD45$^-$EpCAM1$^+$ cells using Sca1, CD63, CD66a, and CD117. **e** Intracellular staining for Dclk1 expression in 4- to 8-week-old mice. Representative histogram and cumulative data depict the percent Dclk1$^+$ cells within tuft-like mTEC and

CD66a$^-$CD117$^-$ non-tuft-like cells, as defined in (**d**) (*n* = 9, from four independent experiments). Statistical analysis was done using a two-tailed unpaired Student's *t*-test. **f**, **g** *Pou2f3*$^{-/-}$ mice were analysed for their abundance of (**f**) Dclk1$^+$ cells and (**g**) CD66a$^+$CD117$^+$ cells compared to C57BL/6 WT mice. Shown are representative FACS plots (left panels) and cumulative data (right panel) (*n* = 6, from two independent experiments). Statistical analysis was done using a two-tailed unpaired Student's *t*-test. **h**, **i** Abundance of tuft-like mTEC, as defined in (**d**) within mTEC$^{lo}$ was analysed at the indicated timepoints in C57BL/6 WT mice. Shown are (**h**) representative FACS plots and (**i**) cumulative data depicting the percent of tuft-like mTEC within TEC and their total cell numbers (W1 *n* = 4, W2 *n* = 10 (percent of TEC) or 8 (number of cells), W4 *n* = 11 (percent of TEC) or 8 (number of cells), W8 *n* = 9, W16 *n* = 11, from 2−3 independent experiments per timepoint). Data are presented as mean +/− SEM in (**e**−**i**). Source data are provided as a Source Data file for panels (**e**−**i**).

TEC (Fig. 5d, f). Clusters 4/D and 5/E related to intertypical TEC while cluster 6/F was most similar to both mature mTEC and proliferating TEC. Clusters 7/G and 8/G displayed a high similarity to mature mTEC (Supplementary Fig. 6f) whereas cluster 9/H linked to tuft-like mTEC. Notably, the transcriptional signatures characteristic of post-Aire mTEC and neural (n) TEC could not be detected. Furthermore, we analysed the data for the expression of gene transcripts characterising the recently described mimetic mTEC subsets (designated tuft, microfold, enterocyte/hepatocyte, neuroendocrine, ciliated, ionocyte, keratinocyte, and muscle cells)[28]. Transcripts identifying "tuft cells" were detected in cluster 9/H (Supplementary Fig. 7) whereas mRNA signatures characterizing microfold-, enterocyte/hepatocyte-, and keratinocyte-like TEC were detected in clusters G/7-8 within the mature mTEC population. Genes associated with neuroendocrine, ciliated, ionocyte, and muscle cells among specified TEC subsets could not be robustly identified.

Replicating the changes of specific TEC subtypes with age, clusters 1/A and 2/B were more abundant in 1-week-old mice whereas the frequencies of clusters 3/C, 4/D, 5/E, 7/G, and 9/H were increased in 16-week-old animals (Fig. 5d; Supplementary Fig. 6g).

TEC in cluster 1/A displayed the highest CD83, CD40, and HVEM protein expression among CITEseq defined clusters, thus confirming the cells' identity as perinatal cTEC. The expression of these markers was reduced in cluster 2/B and completely absent in cluster 3/C, suggesting the former to represent a developmentally intermediate cell state between perinatal and mature/intertypical-like cTEC (Fig. 5d, g). Furthermore, cTEC maturation was paralleled by a decrease in *Foxn1* transcription and an increase in CD73, CD49a, and Sca1 protein expression (Supplementary Fig. 6h).

The differential CD117, CD63, and Sca1 protein expression (as measured by ADT) identified cluster 9/H as tuft-like mTEC (CD117$^+$CD63$^-$Sca1$^-$; see above and Fig. 5d, h) and thus confirmed the flow cytometric definition and gating strategy used to identify these cells as both accurate and practical (Fig. 4g–i). This conclusion was further corroborated by the detection of *Dclk1* and *Ceacam1* transcripts in cluster 9/H (Fig. 5h) and a high similarity score with the tuft-like mTEC subtype (Fig. 5d, f)[13].

ADT-based detection of Sca1 protein expression matched to cells with a transcriptional signature of intertypical TEC within the cTEC (cluster 3/C) and mTEC$^{lo}$ subpopulations (clusters 4 + 5/D +E; Fig. 5d, f, h). Hence, intertypical TEC could unequivocally be identified by Sca1 expression alone. As the transcriptional signature identifying intertypical TEC was spread across three CITEseq-defined clusters (3/C, 4/D and 5/E; Fig. 5d, f), the detection of CD146 expression appeared to deconvolute TEC heterogeneity further since fractions of Sca1$^+$ cTEC and Sca1$^+$ mTEC$^{lo}$ stained positively for CD146$^+$ (Fig. 3e).

The ADT-based documentation of surface markers identified individual TEC subtypes. However, the corresponding gene expression profiles were on their own insufficient to recognize these cells, not least because of the occasional discrepancy between surface protein and RNA expression (Supplementary Fig. 6h, i). CITEseq could

therefore validate the utility of the selected, novel surface markers and the gating strategies chosen. Together they identified four TEC subpopulations that correspond to a specific transcriptionally defined cluster and two subpopulations that represent a mixture of two related clusters, namely UEA1$^-$CD83$^+$CD40$^+$Sca1$^-$ perinatal cTEC (cluster 1/A), UEA1$^-$CD83$^-$CD40$^-$Sca1$^-$ mature cTEC (2/B), UEA1$^-$CD83$^-$CD40$^-$Sca1$^+$ intertypical-like cTEC (3/C), UEA1$^+$MHCII$^{lo}$CD80$^{lo}$Sca1$^+$ intertypical mTEC (4 + 5/D + E), UEA1$^+$MHCII$^{hi}$CD80$^{hi}$ mature mTEC (6 + 7 + 8/F + G), and UEA1$^+$MHCII$^{lo}$CD80$^{lo}$Sca1$^-$CD63$^-$CD66a$^+$CD117$^+$ tuft-like mTEC (9/H) (Supplementary Fig. 8).

## Perinatal cTEC present an enhanced potential for positive selection

We next sought to localize perinatal cTEC within the thymus stromal architecture. Because Ly51 abundance was higher on perinatal cTEC in comparison to other cortical epithelial populations (Fig. 2g), we used this differential to localize perinatal cTEC on thymus tissue sections (Fig. 6a). Quantification of the Ly51 signal intensity in immunohistology detected these cells in close proximity to the medulla with gradual increase of the Ly51 signal but invariable cytokeratin 8 (K8) staining across the cortex from subcapsular region to the inner and eventually deep cortex (Fig. 6a, b).

The newly assigned surface markers enabled the isolation and functional testing of individual cTEC populations ex vivo. We therefore investigated the capacity of perinatal (CD83$^+$CD40$^+$Sca1$^-$) and non-perinatal (CD83$^-$CD40$^-$) cTEC to effect positive thymocyte selection. We co-cultured these cells in thymic epithelial cell cultures (TECx) together with CD69$^-$CD4$^+$CD8$^+$ (i.e., pre-selection double-positive) thymocytes for two days (for gating see Supplementary Fig. 9). Thymocytes were then analysed for phenotypic features associated with positive thymic selection, i.e., the upregulation of TCR and CD5 (Fig. 6c). The number of total thymocytes and those with a CD5$^{hi}$TCRβ$^{hi}$ phenotype were significantly increased in TECx composed of perinatal cTEC when compared to aggregates composed of other cortical epithelia (Fig. 6d). Taken together, these results identified perinatal cTEC to be juxtaposed to the medulla and particularly efficient in positively selecting thymocytes.

The number of perinatal cTEC significantly decreases with age[13]. We therefore explored whether this variation was paralleled by a change in the efficiency to impose positive thymocyte selection. We thus monitored and compared thymocyte maturation in 4- and 16-week-old thymi and classified their sequential maturational stages according to TCRβ and CD69 expression (i.e., stage 0: TCRβ$^-$CD69$^-$ → stage 1: TCRβ$^+$CD69$^{+/-}$ → stage 2: TCRβ$^+$CD69$^+$ → stage 3: TCRβ$^+$CD69$^-$). The frequency of pre-selection thymocytes (i.e., stage 0) was increased, whereas the relative abundance of cells with a post-selection phenotype (stage 2 and 3) was significantly reduced in older animals (Fig. 6e, f). These in vivo results indicated a compromised capacity of older mice to positively select thymocytes, which correlated with a decrease in the availability of perinatal cTEC (Fig. 2e, f).

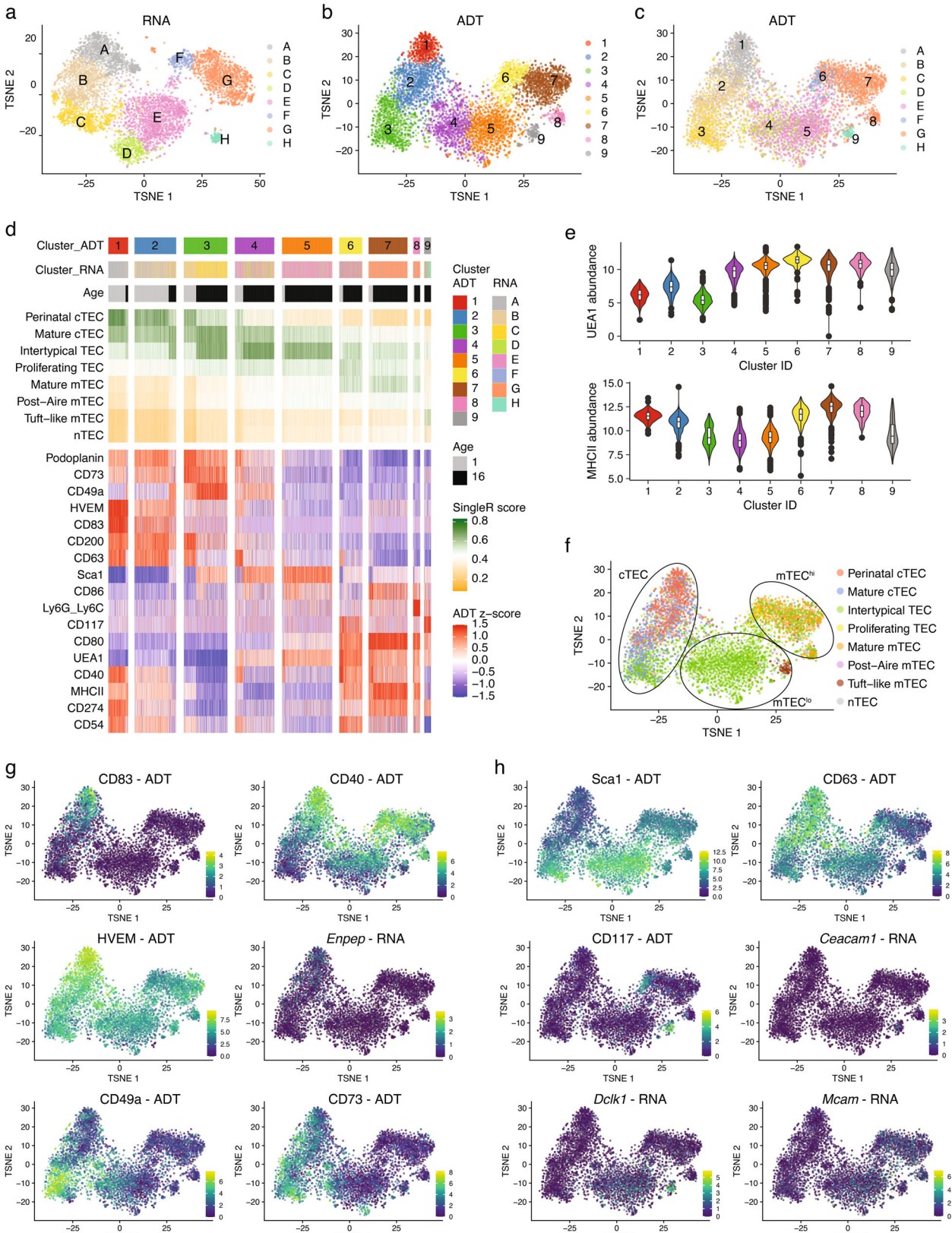

## Crosstalk with thymocytes induces maturation of perinatal cTEC

We finally investigated whether thymic crosstalk[8,29] could explain the inverse correlation between the decreased frequency of perinatal cTEC with age and the expansion of thymocytes after birth. We therefore first determined the frequency of perinatal cTEC in Rag2-deficient

(Rag2[−/−]) mice, which have a hypoplastic thymus secondary to a thymocyte developmental arrest at the DN3a stage. We found a high fraction of perinatal cTEC in these mice that was not influenced by age (Fig. 7a). Hence, thymocytes at developmental stages up to the beta-checkpoint did not influence the age-related changes in perinatal cTEC frequencies.

**Fig. 5 | CITEseq validates new TEC markers.** CD45⁻Ter119⁻ thymic stromal cells isolated from 1- and 16-week-old C57BL/6 WT mice were used for scRNAseq in combination with CITEseq as described in the methods. Cells belonging to clusters assigned as epithelial cells were selected for further analysis. **a–c** Hierarchical clustering analysis was performed on 5834 TEC either using (**a**) the gene expression analysis or (**b**) only considering the detection of ADTs. Results were projected in a 2D space using t-distributed stochastic neighbour embedding (t-SNE). Each colour represents a specific cluster. In (**c**) t-SNE distribution of the ADT clustering is shown using the cluster colouring of the RNA analysis. **d** Compiled data showing the cluster distributions, defined as in a and b, in relation to the derivation of the cells from 1- or 16-week-old mice, and to the similarity score to the reference TEC scRNAseq dataset from Bara-Gale et al[13]. The expression of CITEseq markers is centred to the mean and scaled to the range of expression values. **e** Violin plots depicting the abundance of UEA1 and MHCII ADTs across ADT clusters. The box was drawn from the 25th percentile (Q1) to the 75th percentile (Q3) of the ADT abundance in cells from a specific cluster with the horizontal line denoting the median value. The difference Q3-Q1 forms the interquartile range (IQR). Whiskers are drawn up to the largest data point and down to the smallest data point falling within the range 1.5*IQR. All other observed data points outside the boundary of the whiskers are plotted individually as outliers. **f** Cells were annotated based on transcriptional similarity to the scRNAseq dataset from Baran-Gale et al[13]. Each colour represents a specific TEC subset as defined in the reference dataset. **g**, **h** T-SNE plots illustrating the scaled expression of (**g**) perinatal cTEC markers such as CD83, CD40, HVEM, *Enpep*, CD49a, and CD73, and of (**h**) tuft-like and intertypical TEC markers such as Sca1, CD63, CD117, *Ceacam1*, *Dclk1*, and *Mcam* across ADT clusters.

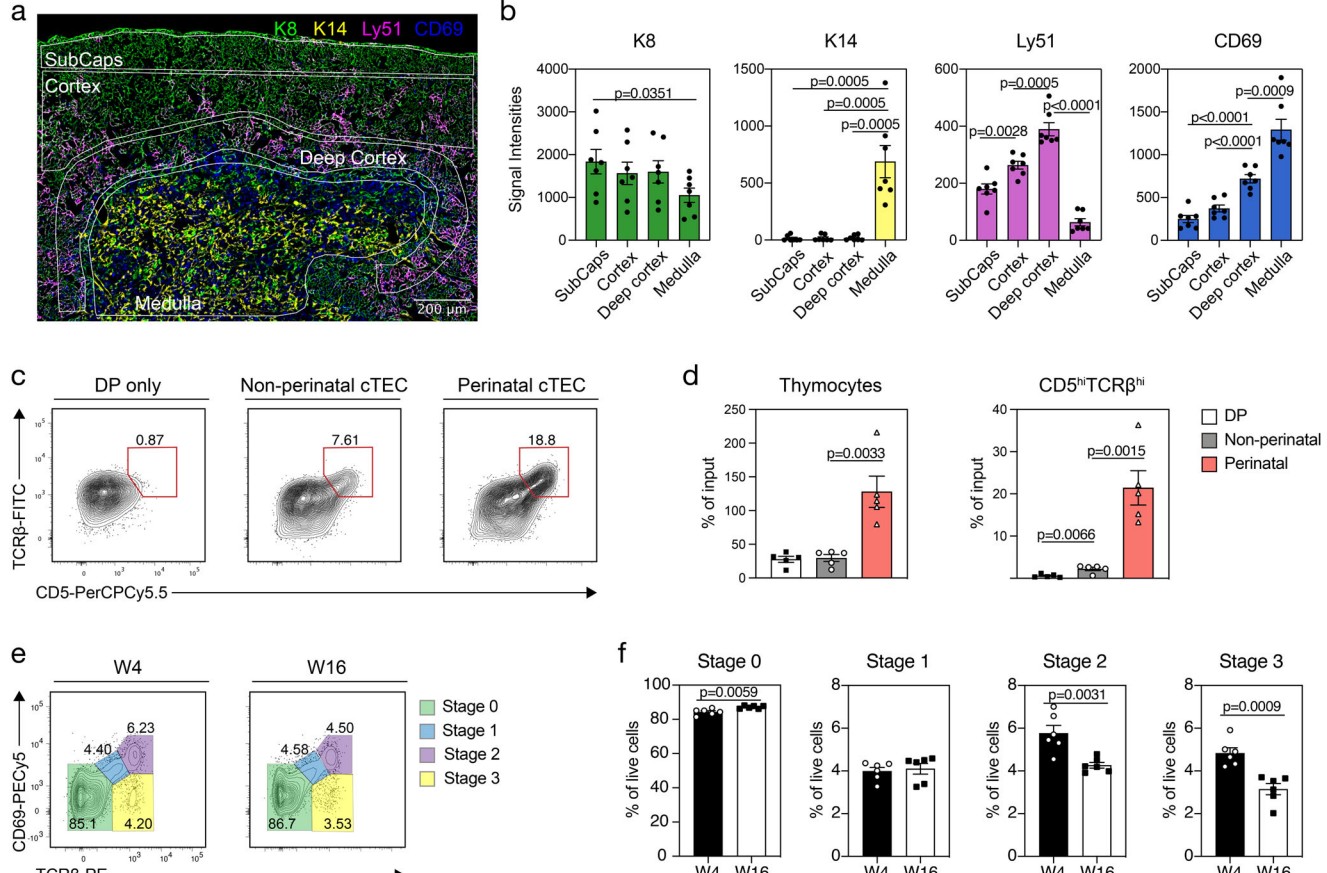

**Fig. 6 | Perinatal cTEC display an increased capacity for positive selection. a**, **b** Immunofluorescent analysis of frozen thymic tissue sections from 4-week-old C57/BL6 WT mice stained with antibodies directed against K8 (green), K14 (yellow), Ly51 (magenta), and CD69 (blue). Shown are (**a**) an image of a representative region (*n* = 7) and (**b**) cumulative data depicting the signal intensities detected across the subcapsular region (SubCaps), the inner cortex, the deep cortex and the medulla. Data are derived from three biological samples. Data are presented as mean values +/− SEM. Statistical analysis was done using a two-tailed unpaired Student's *t*-test. Source data are provided as a Source Data file. **c**, **d** Thymic epithelial cell cultures (TECx) of non-perinatal (CD83⁻CD40⁻) and perinatal (CD83⁺CD40⁺Sca1⁻) cTEC with CD69⁻ DP thymocytes were performed. Shown are representative FACS plots illustrating the expression of (**c**) TCRβ and CD5 after two days of culture for DP only, non-perinatal cTEC and perinatal cTEC cultures and the number of (**d**) thymocytes and CD5ʰⁱTCRβʰⁱ cells acquired (*n* = 5, from three independent experiments). Data are presented as mean values +/− SEM. Statistical analysis was done using a two-tailed unpaired Student's *t*-test. Source data are provided as a Source Data file. **e**, **f** Abundance of developmental thymocyte stages based on the expression of TCRβ and CD69 was analysed in 4- and 16-week-old C57BL/6 WT mice. Shown are (**e**) representative FACS plots and (**f**) cumulative data revealing the percent of cells of thymocyte stages 0–3 (*n* = 6, from two independent experiments). Data are presented as mean values +/− SEM. Statistical analysis was done using a two-tailed unpaired Student's *t*-test. Source data are provided as a Source Data file.

To probe whether thymocytes at later developmental stages, especially unselected CD4⁺CD8⁺ (double positive, DP) thymocytes, controlled the frequency of perinatal TEC, Rag2⁻/⁻ mice were injected with antibodies directed against CD3ε. This treatment results in a substantial increase in pre-selection DP thymocytes[30–32]. Four weeks after antibody or control injections, the thymus of actively treated

Rag2⁻/⁻ mice contained an abundance of DP thymocytes which correlated with numerical and phenotypic changes in the cTEC compartment (Fig. 7b–e). The latter were marked by a reduction in perinatal cTEC, parallel to an increase in non-perinatal cTEC, specifically Sca1⁺ cells (Fig. 7d, e) which corresponded to intertypical TEC according to our CITEseq data (Fig. 5d, f, h). Taken together, these results identified

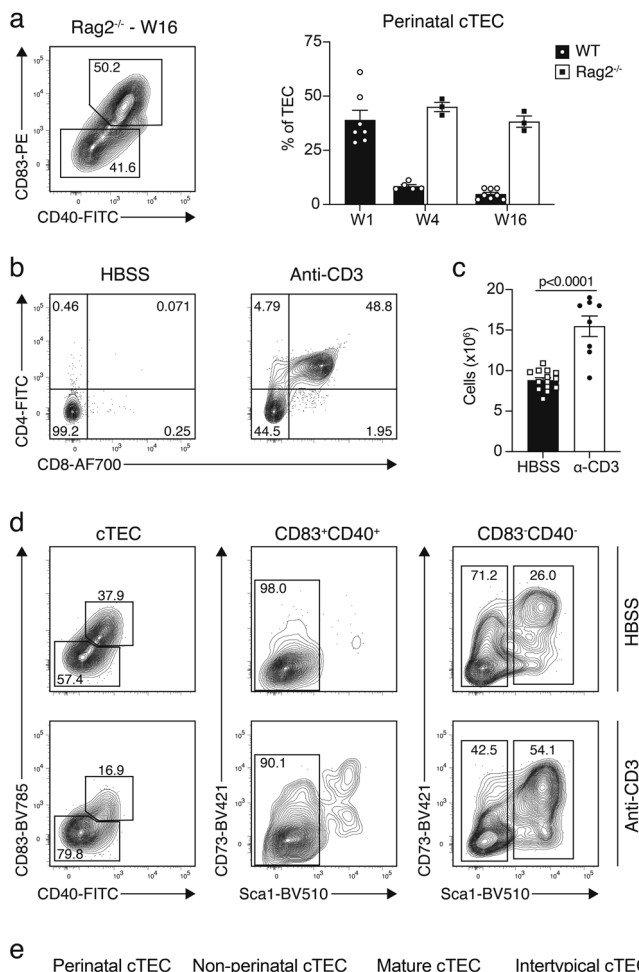

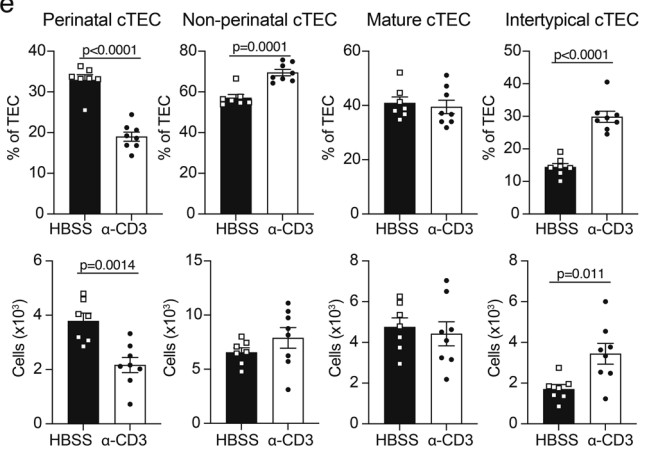

the abundance of and/or signalling by pre-selection DP thymocytes as the mechanism by which the frequency of perinatal cTEC was controlled.

## Discussion

Single-cell transcriptomic analyses have uncovered an unexpected heterogeneity within many cell populations of a seemingly identical phenotype. Cells of the thymic stromal compartment constitute no exception to this observation[4,13–15,22]. The apparent lack of suitable cell surface markers identifying unequivocally TEC subpopulations identical to individual TEC subtypes precludes the isolation of live TEC and their ex vivo functional analysis. Here we report that this limitation has been substantially overcome. We describe novel cell surface markers that identify the comparable subtype of scRNAseq-defined perinatal,

**Fig. 7 | Crosstalk with thymocytes facilitates cTEC maturation. (a)** Rag2[−/−] mice were analysed for the abundance of perinatal cTEC at 4- and 16-weeks and compared to perinatal cTEC in 1-, 4-, and 16-week-old C57BL/6 WT mice. Shown are a representative FACS plots and cumulative data (W1 $n = 7$, W4 WT $n = 5$ and Rag2[−/−] $n = 3$, W16 WT $n = 8$ and Rag2[−/−] $n = 3$). Data shown are derived from one out of two independent experiments. Data are presented as mean values +/− SEM. Source data are provided as a Source Data file. **b, c** Rag2[−/−] mice were injected with HBSS or α-CD3 antibodies (clone KT3) and analysed four weeks later for the development of double positive thymocytes. Shown are (**b**) representative FACS plots depicting the emergence of CD4⁺CD8⁺ cells and (**c**) cumulative data for the total number of cells per thymus (HBSS $n = 14$, α-CD3 $n = 8$, from two independent experiments). Data are presented as mean values +/− SEM. Source data are provided as a Source Data file. **d, e** The cTEC compartment was analysed for changes in the abundance of cTEC subpopulations (CD83⁺CD40⁺Sca1⁻ perinatal, CD83⁻CD40⁻ non-perinatal, CD83⁻CD40⁻Sca1⁻ mature, and CD83⁻CD40⁻Sca1⁺ intertypical cTEC) following α-CD3 treatment. Shown are (**d**) representative FACS plots and cumulative data as percentage of TEC and as total cell numbers (HBSS $n = 7$, α-CD3 $n = 8$, from two impendent experiments). Data are presented as mean values +/− SEM. Statistical analysis was done using a two-tailed unpaired Student's $t$-test. Source data are provided as a Source Data file.

mature and intertypical cTEC, and mature, intertypical and tuft-like mTEC.

Predicting the cell surface phenotypes from corresponding scRNAseq profiles is challenging as technical limitations detecting low transcript copy numbers and the acknowledged disparity between transcript detection and protein expression render this attempt difficult. For example, a comparison of 7 scRNAseq methods uncovered that high-throughput methods, including the widely used 10x Chromium, have lower sensitivities in comparison to the low-throughput methods Smart-seq2 and CEL-Seq2 when capturing rare transcripts[33,34]. We therefore opted for an alternative method and stained TEC for the expression of hundreds of cell surface markers. This screening approach of massive parallel flow cytometry combined with Infinity Flow analysis discovered surface markers previously not inferred to be expressed by TEC. CITEseq which combines the detection of promising candidate markers and single-cell transcriptomic profiles finally established the accuracy of the cell surface markers chosen to identify TEC subtypes.

By applying the newly identified surface markers we show that perinatal cTEC are enriched within the TEC scaffold towards the cortico-medullary junction (CMJ) and that these cells are particularly efficient in positively selecting maturing thymocytes. As expected from their role in shaping the TCR repertoire, we find perinatal cTEC typically juxtaposed to thymocytes with an activated phenotype (i.e., CD69⁺). The age-dependent decline in the frequency of perinatal cTEC is noted both when using flow cytometry and scRNAseq to classify these cells. The actual pace by which this regression is observed differs, however, between these two methods, demonstrating a seemingly faster kinetic for perinatal cTEC identified by their RNA expression profile[13]. This may be explained by differences in the half-lives of specific transcripts and their corresponding proteins. Nonetheless, a post-natal decrease of these cells to an almost complete absence early in adulthood is expected to compromise the robustness of thymopoiesis and possibly the efficiency by which thymocytes are positively selected. The observed decrease in perinatal cTEC correlates with other compositional changes within the epithelial scaffold and may constitute an intrinsic driver for thymus senescence. This understanding is consistent with scRNAseq data that shows the quiescence of a population of medullary precursor cells and correlates this alteration with an impaired maintenance of the medullary TEC compartment[13]. In parallel, these age-related changes link to less efficient T cell selection, a decreased self-antigen representation, a decreased TCR repertoire diversity, and a reduced frequency of thymus-resident naïve T cells (this report and ref. 13).

Intertypical TEC express the glycosyl phosphatidylinositol-anchored cell surface protein Sca1 independently whether they are

positive for the cortical marker Ly51[+] or reactive with the lectin UEA-1, a general feature of medullary TEC. Here, we now show that intertypical TEC can be further split into two sizable subpopulations based on their expression of the cell adhesion molecule CD146. Because oligonucleotide labelled anti-CD146 antibodies were not available for the CITEseq analysis and transcripts for this marker are typically lowly expressed among intertypical TEC, a distinction between CD146 negative and positive cells was not possible when analysing scRNAseq data. Intertypical TEC may contain progenitors with a developmental bias towards the mTEC lineage[13]. Indeed, a recent report provides further support of this contention since the gene expression profile of intertypical TEC largely overlaps with a heterogeneous progenitor population which has been claimed to act as mTEC biased postnatal TEC progenitor[20]. Our profiling of Tspan8 expression and lineage tracing furthermore suggest that the majority of Sca1[+]CD146[+] intertypical TEC relate to immature mTEC that have not yet fully differentiated to express collectively a broad range of tissue restricted antigens.

We further specify the cell surface phenotype for tuft-like mTEC (L1CAM[+]CD104[+])[15]. These cells share transcriptional (e.g. expression of IL25, Trmp5, Dclk1, and IL17RB) and morphological characteristics with gut epithelial tuft cells[13,15,22], play a function in central T cell tolerance induction[22] and control both the homeostasis of type 2 innate lymphoid cells and the generation of type 2 natural killer T cells[15,22,23]. The lineage tracing of these cells further shows that the majority of tuft-like mTEC derive from AIRE expressing TEC, a finding in keeping with previously reported observations[15,22]. However, whether all tuft-like mTEC differentiate from mature mTEC remains an open question because the labelling method needed to draw this conclusion (i.e., inducible, *Aire*-dependent tracing) may not be completely effective[22]. Interestingly, the *Csnb* lineage tracing identified an increased frequency of labelled tuft-like cells in comparison to mature mTEC where labelling is initiated. While we have no unequivocal explanation for this increase, we nevertheless conclude that the majority of tuft-like mTEC (at least 60%) are the progeny of mature medullary epithelia. There is, however, room to speculate that all tuft-like mTEC may be derived in this way, since any contributions from another *Csnb*-negative (i.e., non-labelled) precursor would dilute the frequency of labelled tuft-like cells, a result that we did not observe.

To probe the utility of the new set of cell surface markers to phenotype altered TEC scaffolds, we next analysed the composition of the thymic epithelia in FOXN1[ΔS05/WT] mice which express a dominant negative mutation of FOXN1 and consequently show substantial defects in TEC differentiation[35]. A previous scRNAseq-based analysis of these animals revealed a relative enrichment of perinatal and mature cTEC against a reduction of tuft-like mTEC whereas the frequency of intertypical TEC remained unchanged. The flow cytometric analysis of the TEC scaffold in FOXN1[ΔS05/WT] mice identifies the same variations and thus maps accurately to the transcriptional analysis of these cells, thus demonstrating that the phenotypic and gene expression-based analyses draw comparable conclusions (Supplementary Fig. 10).

With the approach taken, five of the previously defined nine TEC subtypes[13] can now be unequivocally identified using cell surface markers. The still small number of discriminatory cell surface markers so far identified likely accounts for this minor limitation. Implementing more markers in the screening process may identify additional cell surface markers that will identify the remaining TEC subtypes for which we have not yet identified an unambiguous cell surface marker profile. Alternatively, the use of intracellular markers may be informative in identifying the remaining TEC subtypes, namely proliferating TEC, post-AIRE mTEC, and nTEC. However, an obvious drawback for this approach will be that TEC identified in this fashion will be nonviable and can therefore not be used for functional studies in vitro or after transfer in vivo.

It is important to establish precursor-progeny relationships for specific TEC subtypes now that we can identify and purify specific

subpopulations. For example, the potential can be tested whether CD146[+] intertypical TEC give rise to mature mTEC, competent to effect negative selection. Another effort could be directed in dissecting the molecular requirements of tuft-like mTEC controlling the development of type 2 lymphoid cells employing ex vivo functional assays. Finally, the screening workflow described here will also be valuable in identifying novel biomarkers apt to monitor changes in cell subpopulations and their functions resulting from spontaneous or engineered changes in gene function.

Taken together, we have identified novel surface markers that enable the isolation and functional assessment of different TEC subpopulations that correspond to previously identified subtypes so far only defined by their transcriptome. This is accomplished by combining a high throughput screening workflow with a computational expression projection followed by unsupervised clustering.

## Methods

### Mice

Animals were maintained under specific pathogen–free conditions at the University of Oxford Biomedical Science facilities under local and United Kingdom Home Office regulations and permissions. They were housed in cages grouped to a maximum of five animals per cage on wood litter. Environmental enrichment such as tunnel, gnawing and nesting material was provided. The animal house was maintained under artificial lighting (12 h) between 7:00 am and 7:00 pm, in a controlled ambient temperature of 22 °C ± 2 °C, and relative humidity between 30% and 70%. Experiments performed were approved by the United Kingdom Home Office regulations and co-housed age- and gender-matched wild-type C57BL/6 mice were used in all experiments as a reference for genetically modified animals.

*Csnb*[Cre] mice[15] were crossed to the Rosa26[YFP] mouse line[36] to induce lineage tracing in the mature mTEC compartment.

Mice heterozygous for a *Foxn1* allele with a single nucleotide loss at position 1470 (designated FOXN1[ΔS05/WT]) were generated at the Genome Engineering Facility of the MRC Weatherall Institute of Molecular Medicine, University of Oxford as previously described[35].

Rag2[−/−] mice were bred and maintained in the mouse facility of the Department of Biomedicine at the University of Basel in accordance with permissions and regulations of the Cantonal Veterinary Office of Basel-Stadt.

For timed pregnancies 7- to 14-week-old mice were mated overnight and separated early next morning. For pregnant females the mating was considered E0.5 that morning.

### TEC isolation

Isolated thymi were cleaned from adipose tissue, separated into the two lobes, and subsequently subjected to three rounds of enzymatic digestion with Liberase (2.5 mg/ml, Roche, Cat no: 5401127001) and DNaseI (10 mg/ml, Roche, Cat no: 10104159001) diluted in PBS (Gibco, Cat no: 70011044) at 37 °C. After filtration through a 100-μm cell strainer and resuspension in FACS buffer (PBS supplemented with 2% FBS), cell number was determined using a CASY cell counter (Innovatis). For most analyses CD45[+] hematopoietic cells were depleted by incubation with anti-CD45 beads (Miltenyi) as per manufacturer's recommendations and subsequently subjected to the AutoMACS separator (Miltenyi) "depleteS" program.

### Flow cytometry

Cells were counted and stained in FACS buffer containing antibodies of interest for 30 min at 4 °C in the dark. For the identification of dead cells an additional staining with propidium iodide (PI, Sigma, Cat no: P4864) or Zombie red (Biolegend, Cat no: 423110) was used. For intracellular staining, cells were fixed and permeabilised after cell-surface staining using the Cytofix/Cytoperm (BD Biosciences, Cat no: 554714) or the Transcription Factor Staining Buffer Set (Invitrogen, Cat no:

00-5523-00) according to the manufacturer's protocol. Cells were analysed and sorted on a BD FACSAria III instrument (BD Biosciences). Cells were sorted into FACS buffer. Cell purities of at least 95% were confirmed by post-sort analysis. The following antibodies were used: CD4-APCCy7 (1:400, GK1.5, Biolegend), CD4-FITC (1:1000, GK1.5, Biolegend), CD5-PerCPCy5.5 (1:400, 53-7.3, Biolegend), CD8α-AF700 (1:800, 53-6.7, Biolegend), CD40-FITC (1:200, 3/23, Biolegend), CD45-AF700 (1:400, 30-F11, Biolegend), CD63-PE (1:400, NVG-2, Biolegend), CD66a-APC (1:400, Mab-CC1, Biolegend), CD66a-FITC (1:400, Mab-CC1, Biolegend), CD69-PECy5 (1:800, H1.2F3, Biolegend), CD69-PECy7 (1:200, H1.2F3, Biolegend), CD73-BV421 (1:400, TY/11.8, Biolegend), CD80-PECy5 (1:2000, 16-10A1, Biolegend), CD80-BV605 (1:400, 16-10A1, Biolegend), CD83-Bio (1:200, Michel-19, Biolegend), CD83-PE (1:400, Michel-19, Biolegend), CD104-FITC (1:400, 346-11A, Biolegend), CD117-BV421 (1:200, 2B8, BD Biosciences), CD146-APC (1:800, ME-9F1, Biolegend), EpCMA1-PerCPCy5.5 (1:800, G8.8, Biolegend), Dclk1 (1:1000, ab31704, Abcam), HVEM-APC (1:400, HMHV-1B18, Biolegend), Ly51-PECy7 (1:400, 6C3, Biolegend), MHCII-APC/Fire750 (1:2000, M5, Biolegend), UEA1-Cy5 (1:400, Vector Laboratories, in-house labelling), Sca1-BV510 (1:800, D7, Biolegend), TCRβ-FITC (1:400, H57-597, Biolegend), TCRβ-PE (1:1000, H57-597, Biolegend), Tspan8-APC (1:400, FAB6524A, R&D Systems). Biotinylated antibodies were detected using Streptavidin-BV785 (1:500, Biolegend) and unlabelled Dclk1 using anti-rabbit IgG AF647 (1:4000, Invitrogen). Data was analysed using FlowJo (version 10).

## Massively parallel flow cytometry

Cells were isolated and CD45 depletion plus backbone staining were performed as described. The surface backbone panel included antibodies directed against CD45 (1:400, 30-F11, AF700, Biolegend), EpCAM1 (1:800, G8.8, PerCPCy5.5, Biolegend), Ly51 (1:400, 6C3, PECy7, Biolegend), MHCII (1:2000, M5, BV510, Biolegend), CD40 (1:200, 3/23, PECy5, Biolegend), CD80 (1:400, 16-10A1, BV605, Biolegend)), CD86 (1:800, GL-1, BV650, Biolegend), Sca1 (1:1000, D7, BV785, Biolegend), Podoplanin (1:200, 8.1.1, APC, Biolegend), CD31 (1:1000, 390, FITC, Biolegend), the *Ulex europaeus* agglutinin I (UEA1) lectin labelled with biotin (1:1000, Vector Laboratories), followed by secondary streptavidin-BV421 (1:1000, Biolegend) staining and Zombie red (1:1000, Biolegend) staining. Subsequently, the stained cells were distributed across the three 96-well plates provided with the LEGENDScreen kit (Biolegend, Cat no: 700009), each well containing a unique PE-labelled exploratory antibody as well as isotype controls and blanks. PE-labelled antibodies targeting GP2, Tspan8, CD177 and F3 were used as additional exploratory surface antibodies. Due to the low cell numbers obtained after CD45 depletion only ¼ of the recommended quantity of exploratory antibodies was used. Plates were incubated at 4 °C for 30 min in the dark. Thereafter, fixation was performed using the Cytofix buffer (BD Biosciences, Cat no: 554714) for 1 h at 4 °C in the dark. As an additional backbone marker, cells were stained intracellularly for anti-AIRE (1:400, 5H12, AF750, Invitrogen) in Cytoperm buffer (BD Biosciences, Cat no: 554714) and one well stained with anti-FOXN1[37] (1:1600, 2/41, PE, kind gift from Hans-Reimer Rodewald) as an additional exploratory marker, over-night at 4 °C in the dark. The next day cells were resuspended in 100 µl FACS buffer before analysis.

## Infinity Flow and single-cell clustering and expression analysis

For the Infinity Flow computational analysis of the LEGENDScreen datasets, the acquired fcs files were gated on CD45 negative cells or specifically on EpCAM1+ TEC using the FlowJo software. The newly exported fcs files were then used as the dataset for the Infinity Flow pipeline as recently published[21]. The augmented data matrices generated during this process were then further analysed using the Seurat package for hierarchical clustering of the cells and differential expression analysis[24], mostly following the workflow presented in https://satijalab.org/seurat/articles/pbmc3k_tutorial.html. We used

default parameters, except for the data normalization method, normalization.method = "CLR". Values below zero were set to zero to allow for log normalization. Markers were filtered by hand to exclude T-cell related and focus on stromal cell related genes (Supplementary Table 1). The top 20 PCA dimensions were used for clustering and UMAP projections and a clustering resolution of 0.275 was used.

We compared the Infinity Flow data matrices with the scRNAseq dataset of reference [13] by identifying the most closely related genes for each Infinity Flow protein. However, some of the antibodies bind to protein complexes, and here we chose the most abundant transcript related to such a complex—for example, we chose *H2-Ab1* RNA transcripts for MHC class II protein detection. Furthermore, UEA1 detection was identified via *Fut1* RNA expression, since FUT1 synthesises the glycan target of UEA1; for the complete assignment see Supplementary Table 2. We compared the Infinity Flow fluorescence values with the scRNAseq normalised log counts. Clusters from each dataset were then compared using the SingleR package in R[38], with the Wilcox ranked sum test (using the SingleR option de.method = "wilcox").

## Histological analyses

Frozen thymus tissue sections (7 µm) were fixed in acetone and stained using antibodies specific for CD69 (1:100, H1.2F3, Biolegend), Ly51 (1:200, 6C3, Biolegend), K8 (1:500, TROMA-1, NICHD supported Hybridoma Bank), K14 (1:500, Poly19053, Biolegend). Images were acquired using a Leica DMi8 microscope.

## Thymic epithelial cell cultures (TECx)

Perinatal cTEC (CD45−EpCAM1+MHCII+Ly51+CD83+CD40+) and non-perinatal cTEC (CD45−EpCAM1+MHCII+Ly51+CD83−CD40−) were sorted from the thymi of 2-week-old C57BL/6 mice and put in co-cultures with CD69− DP thymocytes sorted from the same thymi, respectively. Cells were transferred in a 1:1 TEC to DP ratio into 1.5 mL tubes containing 1 mL Iscove's modified Dulbecco's medium (IMDM, Gibco, Cat no: 12440053) supplemented with 10% FBS, 100 units/mL penicillin and 100 µg/mL streptomycin (Sigma, Cat no: P4333-100ML) and 1× Gluta-MAX supplement (Gibco, Cat no: 35050061). Co-cultures were maintained at 37 °C in a humidified atmosphere containing 10% $CO_2$ for 48 h and then analysed by FACS. As a control DP cells were also cultured without the addition of TEC.

## Anti-CD3 injections

6- to 7-week-old Rag2−/− animals were injected intraperitoneally with 50ug of anti-CD3ε (clone KT3, in-house produced) or HBSS. Four weeks post injection thymi were analysed for the appearance of DP thymocytes and for changes within their cTEC compartment.

## Cellular indexing of transcriptomes and epitopes by sequencing (CITEseq)

Cells were isolated from six thymi of 1-week- and three thymi of 16-week-old C57BL/6 mice and depleted of CD45+ cells by AutoMACS. Subsequently cells were stained for CD45 (1:400, 30-F11, AF700, Biolegend), EpCAM1 (1:800, G8.8, PerCPCy5.5, Biolegend), Ly51 (1:400, 6C3, PECy7, Biolegend), Ter119-FITC (1:400, TER119, Biolegend) and with PI. In addition cells were stained with antibodies coupled to oligonucleotides directed against CD9 (MZ3, Totalseq-A, Biolegend), CD40 (3/23, Totalseq-A, Biolegend), CD49a (HMα1, Totalseq-A, Biolegend), CD54 (YN1/1.7.4, Totalseq-A, Biolegend), CD63 NVG-2, Totalseq-A, Biolegend), CD73 TY/11.8, Totalseq-A, Biolegend), CD83 (Michel-19, Totalseq-A, Biolegend), CD117 (2B8, Totalseq-A, Biolegend), CD146 (human with cross reactivity to mouse) (P1H12, Totalseq-A, Biolegend), CD200 (OX-90, Totalseq-A, Biolegend), CD274 MIH6, Totalseq-A, Biolegend), HVEM (HMHV-1B18, Totalseq-A, Biolegend), Ly6D (49-H4, Totalseq-A, Biolegend), Ly6C/Ly6G (Gr1) (RB6-8C5, Totalseq-A, Biolegend), MadCAM1 (MECA-367, Totalseq-A, Biolegend), Podoplanin (8.1.1, Totalseq-A, Biolegend), CD80 (16-10A1, Totalseq-A, Biolegend),

CD86 (GL-1, Totalseq-A, Biolegend), MHCII (M5, Totalseq-A, Biolegend), Sca1 (D7, Totalseq-A, Biolegend), CD31 (390, Totalseq-A, Biolegend), EpCAM1 (G8.8, Totalseq-A, Biolegend), CD36 (HM36, Totalseq-A, Biolegend), CD133 (15-2C11, Totalseq-A, Biolegend), CD157 BP-3, Totalseq-A, Biolegend), CD300LG (ZAQ5, Totalseq-A, Biolegend), and the *Ulex europaeus* agglutinin I (UEA1) lectin labelled with biotin, followed by secondary staining with streptavidin-PE (Totalseq-A, Biolegend) coupled to an oligonucleotide. CD45⁻Ter119⁻EpCAM1⁺ and CD45⁻Ter119⁻EpCAM1⁻ cells were sorted in a 70–30% ratio into a 1.5 mL tube containing FACS buffer for the 1-week-old and 16-week-old samples, respectively. For both timepoints an estimate of 28,000 total cells were loaded on two wells of a 10x Genomics Chromium Single Cell Controller. After single-cell capture cDNA and library preparation were performed according to the manufacturer's instructions using a Single-Cell 3' v3 Reagent Kit (10x Genomics) with the changes as described in[26] to capture cDNA and produce libraries from antibody-derived oligos (ADT). Sequencing was performed on one lane of the Illumina NovaSeq 6000 system with a mix of 90% cDNA library and 10% ADT library resulting in 151nt-long paired-end reads.

The dataset was analysed by the Bioinformatics Core Facility, Department of Biomedicine, University of Basel. cDNA reads were aligned to 'mm10' genome using Ensembl 102 gene models with the STARsolo tool (v2.7.9a) with default parameter values except the following parameters: soloUMIlen=12, soloBarcodeReadLength=0, clipAdapterType=CellRanger4, outFilterType=BySJout, outFilterMultimapNmax=10, outSAMmultNmax=1, soloType=CB_UMI_Simple, outFilterScoreMin=30, soloCBmatchWLtype=1MM_multi_Nbase_pseudocounts, soloUMIfiltering=MultiGeneUMI_CR, soloUMIdedup=1MM_CR, soloCellFilter=None. ADT libraries were also processed using the STARsolo tool with default parameters except soloCBmatchWLtype=1MM_multi_Nbase_pseudocounts, soloUMIfiltering= MultiGeneUMI_CR, soloUMIdedup=1MM_CR, soloCellFilter=None, clipAdapterType=False, soloType=CB_UMI_Simple, soloBarcodeReadLength=0, soloUMIlen=12, clip3pNbases = 136.

Further analysis steps were performed using R (v4.1.2). Note that cell filtering was done based only on the analysis of the gene expression, not ADT abundance. Cells were considered as high-quality cells if they had at least 2000 UMI counts, which is the threshold derived from the distribution of UMI counts across cells, forming a data set of 9953 cells.

Multiple Bioconductor (v3.14) packages including DropletUtils (v1.14.2), scDblFinder (v1.8.0), scran (v1.22.1), scater (v1.22.0), scuttle (1.4.0) and batchelor (v1.10.0) were applied for the further analysis of the dataset mostly following the steps of the workflow presented at https://bioconductor.org/books/release/OSCA/. Normalised[39] log-count values for the gene expression were used to construct a shared nearest-neighbour graph[40], which nodes, i.e., cells, were clustered by 'cluster_louvain' method from the R igraph package[41]. Counts reflecting the ADT abundance in cells were also log-normalised and clustered in a similar manner.

The data set was subjected to the cell-type annotation using the Bioconductor package SingleR (v1.8.1) and samples from the Immunological Genome Project (ImmGen) provided by the Bioconductor package celldex (v1.4.0) as the reference. Clusters of cells mostly assigned to 'Epithelial cells' (5834 cells) were filtered (Supplementary Fig. 6d). Note that one of the clusters (cluster A, Supplementary Fig. 6a) was excluded at this step, because it was mostly composed of cells with elevated percentage of reads mapping to mitochondrial and ribosomal genes and lower number of counts.

The gene expression of filtered cells was re-analysed by removing the batch effect formed by the combination of the sample of origin and the number of counts per cell (cells with >12,000 counts and cells with 12,000 counts) and re-clustered (Fig. 5a, b). Cells were also subjected to the cell-type annotation using scRNAseq transcriptional profiles of single TEC as the reference data set[13] (Fig. 5d, f). The scoreMarkers

function of the scran package was applied to find marker genes of clusters 1–3. The standardised log-fold change across all pairwise comparisons 'mean.logFC.cohen'>1 was used as the significance threshold defining the set of marker genes.

A t-SNE dimensionality reduction was used for visualizing single cells on two dimensions. T-SNE coordinates were calculated using the runTSNE function from the scater package and default parameters. For the visualization of cells based on the gene expression, coordinates of principal components and 2000 most variable genes with excluded mitochondrial and ribosomal genes were used as the input. For the visualization of cells based on the ADT abundance, coordinates of principal components and all ADTs were used as the input.

## Bulk RNA sequencing

Triplicates of Sca1⁻CD63⁻CD66a⁺CD117⁺ tuft-like and Sca1⁻CD63⁻CD66a⁻CD117⁻ non-tuft-like mTEC cells were sorted from thymi of 6-week-old C57BL/6 mice into trizol (Invitrogen, 15596026). Subsequently samples were submitted to ultra-low input bulk RNAseq. Reads were trimmed using Trimmomatic (version 0.36) to remove adapter sequences and aligned to the mouse genome (mm10) using STAR (version 2.7.3a)[42,43]. HTSeq (version 0.12.4) was used to assign reads to genes with the option "intersection-nonempty"[44]. Differentially expressed genes were identified using edgeR (FDR < 0.05)[45]. Spearman correlation coefficients were calculated between Sca1⁻CD63⁻CD66a⁻CD117⁻ non-tuft-like mTEC bulk RNAseq samples and scRNAseq data from reference [13] for all differentially expressed genes with $\log_2$ fold change ≥1.

## Statistical analysis

GraphPad Prism (version 9) was used to perform all statistical analyses, except for the bulk RNAseq and CITEseq datasets. The statistical tests used are described in the figure legends, and exact p-values are shown within each figure. Non-significant differences are not specified.

## Reporting summary

Further information on research design is available in the Nature Portfolio Reporting Summary linked to this article.

## Data availability

The CITEseq and the bulk RNAseq datasets have been deposited in the Gene Expression Omnibus database under accession numbers GSE215418 and GSE226128, respectively. Source data are provided with this paper.

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

## Acknowledgements

We thank Emilie Cosway and Sonia Parnell for their technical assistance and Lilly von Muenchow for critical reading of the manuscript. Further we would like to acknowledge Grozdan Cvijetic for his help to set up the Infinity Flow computational analysis pipeline. We also thank Jakub Abramson for sharing the *Csnb*^Cre mice. This research was supported in part by the generosity of an anonymous donor to Stanford University establishing the DiGeorge Syndrome Research Fund (G.A.H.), Swiss National Science Foundation grant IZLJZ3_171050; 310030_184672 (G.A.H.), Medical Research Council grant MR/S036407/1 (G.A.H.), Wellcome Trust grant 105045/Z/14/Z and 211944/Z/18/Z (G.A.H.); Swiss National Science Foundation Early Postdoc.Mobility Fellowship (P2BSP3_188183) and Postdoc.Mobility Fellowship (P500PB_206823) to F.K.; NIHR Clinical Lectureship to F.D.; work in the G.A. lab is supported by an MRC Programme Grant (MR/T029765/1).

## Author contributions

F.K., C.V.-V., S.M., F.D., S.Z., L.M., I.C.-A., M.E.D., A.J.W., and B.L. performed experiments; F.K., C.V.-V., S.P., A.B., F.D., A.E.H., S.Z., and L.M. analysed data; F.D. provided CsnbCre mice; G.A. provided the thymi from Pou2f3-/- mice; F.K. and G.A.H. conceived the project and wrote the manuscript.

## Competing interests

The authors declare no competing interests.
