## [Peer Review File · Nature Communications]

REVIEWER COMMENTS

Reviewer #1 (expert in mTEC, AIRE, and transcriptional regulation of T cell development):

The manuscript of Kevin and al. draw attention to the complexity and heterogeneity of thymic epithelial cells, especially with the identification of perinatal cTEC and their impact on the positive selection of immature thymocytes. The screening of surface markers identifies novel targets that greatly facilitate the isolation of specific TEC populations and therefore, establish precursor-progeny relationships. Overall, the experiments and the bioinformatics analyses are well performed and provide interesting information on TEC diversity and their discriminatory surface markers. However, the current manuscript would benefit from additional information:

1) Lines 101-105: "At each of the three separate time points, the major thymic stromal cell types, epithelia, fibroblasts, pericytes, and endothelial cells, could reliably be identified based on the expression of key markers including EpCAM1 (CD326) identifying TEC, CD140a and Podoplanin marking fibroblasts, Ly51 and CD146 singling out pericytes, and CD31 staining endothelial cells. Additional markers identified subsets within these cell populations (see Figures S1A-C)." The classification of Pdpn as a key marker for fibroblasts doesn't seem accurate. Indeed, as shown in the Infinity Flow analysis, there is a higher expression of Pdpn in cTEC than in fibroblasts (FigS1). Pdpn also seems to be a marker of intertypical TEC as stated in the introduction (line 64).

2) Lines 113-115: "In a second analysis we focused exclusively on EpCAM1+ cells and disclosed in 1-week-old but not older mice three separate cTEC subclusters as defined by the cells' differential expression of Ly51, UEA1, MHCII, and CD80". In Fig 1B, since UEA-1 and CD80 are not expressed in cTEC and since Ly51 is absent from mTEC clusters but expressed at different levels in each of the 3 cTEC clusters, it seems that Ly51 alone is sufficient to discriminate cTEC identity?

3) The lack of immunostaining pattern of antibodies FIG6A directed against the perinatal cTEC key markers (CD83, CD40, HVEM) undermines their spatial identification. An additional immunostaining with another marker such as CD124 (cf Fig1B) could be used to support Ly51 staining.

4) The RTOC co-culture provides interesting insights into the perinatal cTEC impact on T cell development. However, RTOC is a technique based on cell reaggregation in an organoid/ spheroid structure. It doesn't seem to be the case here as written in the Methods section, "Reaggregate thymic organ cultures"?

5) Perinatal cTEC defined by CD83 expression have an impact on T cell development. The correlation of the postnatal decrease of these cells and the compromised capacity of older mice to positively select thymocytes is really interesting. Similarly, Fujimoto and al. (2021) showed that CD83+ dendritic cells influence T cell development. Did the authors check CD83+ dendritic cells in their dataset to see if there is a decrease of this population too?

6) In Fig1, Sca1 CD80 CD40 Ly51 are present in duplicate. Does it correspond to antibodies recognizing different specificities of these proteins?

7) Fig 5A-C would benefit from text labels to identify the clusters.

Reviewer #2 (expert in massively parallel flow cytometry):

Summary: This study by Klein et al. utilizes massively parallel flow cytometry, Infinity Flow analysis, and CITE-seq to define heterogeneity in Murine Thymic Stromal cell populations at 2 developmental ages (PND1, PND4, and PND16.) Through these studies the authors establish a cell surface expression profile across the entire range of thymic stromal subtypes, map alterations in these populations over development, and identify previously unknown heterogeneity within both cTEC and mTEC populations. These findings were validated through comparison to a robust

CITEseq dataset which allows comparison of a subset of surface markers as well as transcriptional data. Using these data the author develop flow cytometric gating strategies to enable sorting of the newly defined subsets and assess function in ex vivo assays. With respect to this review – I've limited my comments specifically to the aspects of the study involving Infinity Flow and high-dimensional flow cytometry which fall within my areas of expertise- I will not comment on the aspects of thymic development as I do not have proper background in this area in order to rigorously assess the findings.

1) All aspects of the massively parallel flow cytometry study have been implemented well and are generally well described. However, a detailed description of the Backbone panel utilized for this assay (inclusive of specific antibody clones and fluorophores) should be provided to enable clear assessment of the resulting data.

2) Additional detail should be provided as to how Seurat was utilized for clustering and DE based analysis – specifically whether any additional normalization or transformation of the data was performed as part of this analysis. Depth similar to what is provided for the CITEseq analysis should be provided.

3) The authors utilize a clever similarity score-based approach to cross-compare CITEseq and Infinity Flow generated datasets. Additional details need to be provided to fully assess this – specifically it is noted that in some cases direct correlates for antibody staining and transcript were not possible – such that the authors utilized the closest approximation (Fut1 mRNA for UEA1 antibody staining is the example given.) I don't see any specific issue with doing this however all instances where this was done should be documented specifically in a table.

With these minor issues addressed the data presented are of sufficient rigor to support the analysis as presented. Presuming the biology focused aspects of this study are similarly meritorious I would consider this study acceptable for publication.

Reviewer #3 (expert in mTEC and transcriptomic analysis of mTEC):

The manuscript by Klein et al utilizes massively parallel flow cytometry followed by computational machine learning algorithm to identify surface markers specific to various cell subsets of thymic stroma.

Although the study offers very limited insights into the putative functional roles of the individual stromal subsets, it provides a very useful molecular resource for future studies in the field, enabling a more precise isolation and characterization of the individual cell subsets. Importantly, the study also highlights major cellular and molecular differences between the neonatal, adult and aging thymus.

Overall, the study is well performed and written, and is potentially suitable for publication in Nat Comm as a resource. however, I suggest the authors try to address the following points to further improve the clarity and presentation of their findings

The coloring in Fig 1B-D, which show TEC heterogeneity at different ages is rather confusing. It is clear that the stromal composition is not the same at each timepoint, however, for the sake of clarity, the authors should try to keep the colors for the main subsets and their analogues uniformed. For instance in W1 cTECs are divided into 3 subsets marked by orange, green and brown, while at other ages, green and brown are used for subsets of mTECs. This is highly confusing. I suggest the authors use different gradients of the same color for the major cell subsets (cTEC, mTEChi, early mTEClo, mTEClo late). Similarly, the authors should try to keep the same order of genes in B,C, D, as much as possible. For instance, Ly51 is on first, third and second position in 1B, C, D respectively. All backbone markers should be shown for all ages.

The authors suggest the existence of 3 unique cTEC subsets in the neonatal thymus. The difference between cTEC2 and 3 is however not clearly apparent. The data rather argue for the existence of two major cTEC subsets that can be distinguished by differential expression of surface markers.

The identification of CD66a and CD117 as novel markers of thymic tuft cells is a very interesting finding. These should however be validated further in more detail

- The authors should provide a better characterization of the Sca1- CD63- Cd117- CD66a- population of non-tuft cells in Fig 4D – this population represents almost 40% of the Sca1- CD63- population and should be characterized further (e.g. by bulk RNAseq; or qPCR or FACS, to identify what is the key gene signature that defines this subsets)

- Similarly the authors should provide a better characterization of the CD117+ CD66a+ Dcl1- population – are these really tuft cells that do not express Dcl1 or does this subset contain other post Aire subsets? Does these cells express other tuft cells markers, such as L1CAM?

- Several previous studies suggested that there is internal heterogeneity in the tuft cell compartment, can the authors differentiate between tuft cell subsets using the newly identified surface markers

- Which of the adult mTEC populations is enriched for other “mimetic” cells subsets, such as keratinocyte-like TECs, (neuro)endocrineTECs, myoTEC, etc. these recently described rare subsets should be mapped

most of these points could be addressed by using the proposed gating strategies and validating the corresponding gene signatures by bulkRNAseq.

RESPONSE TO REVIEWERS' COMMENTS

Reviewer #1

1) Lines 101-105: *“At each of the three separate time points, the major thymic stromal cell types, epithelia, fibroblasts, pericytes, and endothelial cells, could reliably be identified based on the expression of key markers including EpCAM1 (CD326) identifying TEC, CD140a and Podoplanin marking fibroblasts, Ly51 and CD146 singling out pericytes, and CD31 staining endothelial cells. Additional markers identified subsets within these cell populations (see Figures S1A-C).”* The classification of Pdpn as a key marker for fibroblasts doesn't seem accurate. Indeed, as shown in the Infinity Flow analysis, there is a higher expression of Pdpn in cTEC than in fibroblasts (FigS1). Pdpn also seems to be a marker of intertypical TEC as stated in the introduction (line 64).

The reviewer is correct in her/his observation that staining for Podoplanin does not uniquely identify fibroblasts. We have therefore removed the corresponding statement in the manuscript as its wording is not entirely accurate and could be misunderstood that podoplanin-positivity exclusively marks fibroblasts. The revised passage now reads: “...CD140a marking fibroblasts, ...” [line 115].

2) Lines 113-115: *“In a second analysis we focused exclusively on EpCAM1+ cells and disclosed in 1-week-old but not older mice three separate cTEC subclusters as defined by the cells' differential expression of Ly51, UEA1, MHCII, and CD80”. In Fig 1B, since UEA-1 and CD80 are not expressed in cTEC and since Ly51 is absent from mTEC clusters but expressed at different levels in each of the 3 cTEC clusters, it seems that Ly51 alone is sufficient to discriminate cTEC identity?*

We appreciate the reviewer's observations and fully agree with her/his remark that Ly51, the well-established cTEC cell surface marker, is detected on all cortical but is absent from medullary epithelial cell clusters. However, we noted that only the cTEC I population expresses high levels of Ly51 (see Figure 1B). Moreover, phenotyping individual TEC subpopulations throughout the life course of the mouse demonstrates that the expression of Ly51 varies considerably across the life-course, yielding especially in older mice populations that contain both mature cTEC and Intertypical TEC. Hence, the cTEC phenotyping employing flow cytometry benefits from the combined use of UEA1 (absent) and Ly51 (variable) as exemplified in 16-week-old animals (see Figure 4D and Figure S8). We agree with the reviewer's valuable comment that CD80 and MHCII expression are not required for the unequivocal identification of cTEC. We have therefore edited the corresponding statement which now reads: “.... three separate cTEC subclusters as defined by the cells' differential expression of Ly51 and UEA1, thus illustrating a greater heterogeneity of the cTEC population early in life (Figure 1B-D and Figure S2A-C).” line 126-132].

3) *The lack of immunostaining pattern of antibodies FIG6A directed against the perinatal cTEC key markers (CD83, CD40, HVEM) undermines their spatial identification. An additional immunostaining with another marker such as CD124 (cf Fig1B) could be used to support Ly51 staining.*

We thank the reviewer for her/his helpful comment and concur that the use of an additional marker would affirm the immunostaining pattern of perinatal cTEC. For the evaluation and selection of additional markers identifying this cTEC population we scrutinized data from both our flow cytometry and scRNAseq analyses. Perinatal cTEC expressed CD83, CD40, and Ly51 protein and their matching transcripts whereas the HVEM (CD270) protein was identified on the cell surface by flow cytometry, but the matching transcripts could not be detected in perinatal cTEC at single cell resolution. Moreover, only the detection of Ly51 provided an informative immunohistochemical pattern thus precluding an unambiguous spatial resolution identifying perinatal cTEC within the thymic microenvironment. Other proteins including CD124 were omitted as markers to identify perinatal cTEC because there was a lack in concordance between the protein expression and the detection of the respective transcripts, low expression in cTEC by flow cytometry, and/or co-expression by thymocytes.

Expression of CD124 (encoded by *Il4ra*): (A) Left: Hierarchical clustering analysis was performed on single-cell RNA-sequencing data obtained from TEC isolated from 1-, 4-, 16-, 32, and 52-week-old mice and projected in a 2-dimensional space using UMAP. Right: UMAP graph illustrating the scaled expression of *Il4ra*. Colour gradient indicates the relative expression level. (B) FACS plot showing the surface expression pattern of CD124 in cTEC (UEA1-) and mTEC (UEA1+) derived from 1-week-old mice.

4) *The RTOC co-culture provides interesting insights into the perinatal cTEC impact on T cell development. However, RTOC is a technique based on cell reaggregation in an organoid/ spheroid structure. It doesn't seem to be the case here as written in the Methods section, "Reaggregate thymic organ cultures"?*

We thank the reviewer for this comment and agree with her/his point. As a result, we have renamed the method as "Thymic Epithelial Cell Cultures" (TECx).

5) *Perinatal cTEC defined by CD83 expression have an impact on T cell development. The correlation of the postnatal decrease of these cells and the compromised capacity of older mice to positively select thymocytes is really interesting. Similarly, Fujimoto and al. (2021) showed that CD83+ dendritic cells influence T cell development. Did the authors check CD83+ dendritic cells in their dataset to see if there is a decrease of this population too?*

We appreciate the reviewer's comment and the reference to work by Y. Fujimoto and colleagues (2002) that demonstrated CD83 to be expressed by both mouse thymic dendritic cells (DC) and epithelial cells and that further showed that CD83-deficient (CD83^{-/-}) mice had a specific block in CD4⁺ single-positive (SP) thymocyte development. The transfer of wild-type epithelial cells but not wild type DC into thymi of CD83^{-/-} mice increased CD4⁺ SP T cell production by 2-fold demonstrating that epithelial CD83 expression contributes substantially to this maturation. As the focus of our work presented here was to identify novel TEC markers that phenotypically specify subtypes so far only defined by their transcriptome, we did not probe the expression of CD83 on thymic DCs over an animal's life course. However, we determined the change in the relative frequency of CD83-positive DC in 4 and 14-week-old mice, respectively. Although the total thymic cellularity was decreased between these two time points in older mice (left bar graph, below) the relative frequency of CD83+ DC did not change as a function of age and contrast the reduction of perinatal cTEC in the same time period.

CD83+ DC in 4- and 14-week-old mice: The abundance of CD83+ DC (CD11c^{hi}MHCII^{hi}) was determined in 4- and 16-week-old mice by flow cytometry. Bar graphs show total thymic cellularity (left graph), percent of CD83+ DC as a relative frequency of all cells (middle graph), and the absolute cellularity of thymic CD83+ DC at the indicated ages.

6) *In Fig1, Sca1 CD80 CD40 Ly51 are present in duplicate. Does it correspond to antibodies recognizing different specificities of these proteins?*

The reviewer's queries why some antibodies were apparently used twice for the analysis shown in Figure 1. Antibodies recognizing Sca1, CD80, CD40, and Ly51 were used as backbone markers to unequivocally identify specific TEC subpopulations using "standard cell surface markers" but were also included in the set of exploratory markers to identify novel markers as internal control. To make this distinction clear, we have added in the revised Figure 1 the designation "BB" to highlight the backbone markers and accordingly changed their font color to blue.

7) *Fig 5A-C would benefit from text labels to identify the clusters.*

We agree with the reviewer's comment and have now added text labels to the Figure panels.

Reviewer #2

1) *All aspects of the massively parallel flow cytometry study have been implemented well and are generally well described. However, a detailed description of the Backbone panel utilized for this assay (inclusive of specific antibody clones and fluorophores) should be provided to enable clear assessment of the resulting data.*

In response to the reviewer's valuable comment, the revised manuscript now details the requested information concerning antibody clones and used fluorochromes (Lines 589-592).

2) *Additional detail should be provided as to how Seurat was utilized for clustering and DE based analysis – specifically whether any additional normalization or transformation of the data was performed as part of this analysis. Depth similar to what is provided for the CITEseq analysis should be provided.*

We appreciate the reviewer's point and have therefore edited the corresponding paragraphs in the Methods section (Lines 611-627). We believe that the section entitled "Infinity Flow and single-cell clustering and expression analysis" now contains all the necessary information.

3) *The authors utilize a clever similarity score-based approach to cross-compare CITEseq and Infinity Flow generated datasets. Additional details need to be provided to fully assess this – specifically it is noted that in some cases direct correlates for antibody staining and transcript were not possible – such that the authors utilized the closest approximation (Fut1 mRNA for UEA1 antibody staining is the example given.) I don't see any specific issue with doing this however all instances where this was done should be documented specifically in a table.*

We welcome the reviewer's comment and provide detailed information about the correlation of antibody staining and transcripts in a new, separate table in the revised manuscript (Supplementary Table 2; page 22-24 of supplementary materials).

Reviewer #3 (expert in mTEC and transcriptomic analysis of mTEC):

1) *The coloring in Fig 1B-D, which show TEC heterogeneity at different ages is rather confusing. It is clear that the stromal composition is not the same at each timepoint, however, for the sake of clarity, the authors should try to keep the colors for the main subsets and their analogues uniformed. For*

instance in W1 cTECs are divided into 3 subsets marked by orange, green and brown, while at other ages, green and brown are used for subsets of mTECs. This is highly confusing. I suggest the authors use different gradients of the same color for the major cell subsets (cTEC, mTEC^{hi}, early mTEC^{lo}, mTEC^{lo} late). Similarly, the authors should try to keep the same order of genes in B,C, D, as much as possible. For instance, Ly51 is on first, third and second position in 1B, C, D respectively. All backbone markers should be shown for all ages.

We thank the reviewer for this helpful comment. In response, we changed the color code of the TEC clusters according to the reviewer's suggestion, and the cTEC, mTEC^{lo} and mTEC^{hi} clusters have now the same colors across the three timepoints analyzed. However, the ranking of the genes in panels B-D in Figure 1 has been accomplished in an unsupervised manner and can therefore not be implicitly changed. The revised Figure S2 now also shows the backbone markers for all ages in the same order.

2) The authors suggest the existence of 3 unique cTEC subsets in the neonatal thymus. The difference between cTEC2 and 3 is however not clearly apparent. The data rather argue for the existence of two major cTEC subsets that can be distinguished by differential expression of surface markers.

We appreciate the reviewer's observation and agree with her/his remark. Clusters cTEC2 and cTEC3 are indeed very similar, although an unsupervised computational analysis concluded that these two clusters are separate. Nonetheless and in response to the reviewer's comment, we have focused our analysis to describe differences between cTEC1 cluster and the combined cluster of cTEC2 and cTEC3.

3) The identification of CD66a and CD117 as novel markers of thymic tuft cells is a very interesting finding. These should however be validated further in more detail
3.1) The authors should provide a better characterization of the Sca1- CD63- Cd117- CD66a- population of non-tuft cells in Fig 4D – this population represents almost 40% of the Sca1- CD63- population and should be characterized further (e.g. by bulk RNAseq; or qPCR or FACS, to identify what is the key gene signature that defines this subsets)

We thank the reviewer for his/her helpful comment. In response, we performed bulk RNAseq of thymic “non-tuft-like” (Sca1-CD63-CD66a-CD117- mTEC^{lo}) vs. tuft-like cells (Sca1-CD63-CD66a+CD117+ mTEC^{lo}) and compared the cells' gene expression profiles. The gene expression of the latter cells showed a prominent expression of *Ceacam1*, *Kit*, *Dclk1*, and *Il25* and other genes characteristically hallmarking tuft-like cells as recently demonstrated by Baran-Gale and colleagues (Baran-Gale, J. *et al.* Ageing compromises mouse thymus function and remodels epithelial cell differentiation. *Elife* 9 (2020)). In contrast, the “non-tuft-like” cells displayed a substantially different gene expression profile lacking the expression of the top 20 transcripts typically identifying tuft-like mTEC. Thus, the bulk RNAseq data confirms that Sca1-CD63-CD66a+CD117+ cells display a tuft-like mTEC signature. The gene expression signature of “non-tuft-like” mTEC did not match any of the TEC subtypes as previously defined by Baran-Gale *et al.*. We therefore infer that these “non-tuft-like” cells likely represent a mixture of different mTEC subpopulations. This information is now included in the revised manuscript (see lines 218-230 and Figure S4G-I).

3.2) Similarly the authors should provide a better characterization of the CD117+ CD66a+ Dclk1- population – are these really tuft cells that do not express or does this subset contain other post Aire subsets? Does these cells express other tuft cells markers, such as LICAM?

We thank the reviewer for this question and agree that it is not entirely clear if the Dclk1- fraction encompasses tuft-like mTEC. We therefore sought to analyze the expression of L1CAM and CD104 in both the Dclk1-positive and -negative subsets captured as tuft-like mTEC. Despite several attempts and the use of different approaches we were unable to reliably stain for L1CAM and thus could not confirm the staining reported by Bornstein and colleagues (Bornstein, C. *et al.* Single-cell mapping of the thymic stroma identifies IL-25-producing tuft epithelial cells. *Nature* 559, 622-626 (2018)). The staining for CD104 revealed a low cell surface expression on both Dclk1-positive and -negative tuft-

like mTEC when compared to intertypical TEC (as previously observed by Bornstein *et al*). To determine whether tuft-like mTEC also contain *Dclk1*-negative cells, we analyzed the scRNAseq dataset reported by Baran-Gale et al for the expression of *Dclk1*. Out of 132 cells assigned to the tuft-like mTEC cluster, 71 were negative for *Dclk1* (panel A, figure below). Moreover, scRNAseq data from one of the original publications describing tuft-like mTEC demonstrated in Figure 2f an absence of *Dclk1* transcripts in a substantial fraction of the cells (Miller, C.N. *et al*. Thymic tuft cells promote an IL-4-enriched medulla and shape thymocyte development. *Nature* **559**, 627-631 (2018). This information as well as the CD104 staining is now included in the revised manuscript (see Figure S4D and lines 207-210).

***Dclk1* negative tuft-like mTEC:** Violin plot illustrating the expression of *Dclk1* within the tuft-like mTEC cluster based on data from Baran-Gale et al.

3.3) Several previous studies suggested that there is internal heterogeneity in the tuft cell compartment, can the authors differentiate between tuft cell subsets using the newly identified surface markers

We agree with the reviewer's comment that a further classification of the heterogeneity of tuft-like cells using novel cell surface markers would be welcome as they could characterize these cells further, possibly identifying additional subsets. Neither a re-analysis of the CITEseq data from tuft-like mTEC using tSNE nor a heat map displaying the expression levels of cell surface markers used for the CITEseq identified separate subclusters. As a caveat we may however like to add that the low number of tuft-like mTEC in this analysis precluded a robust identification of separate tuft-like mTEC subpopulations, should they indeed exist.

Heterogeneity within tuft-like mTEC: (A) Re-clustering of cells annotated as tuft-like mTEC (cluster H) analysing CITEseq data. **(B)** Heatmap depicting the expression of all surface markers employed for the CITEseq analysis of tuft-like mTEC (cluster H).

3.4) Which of the adult mTEC populations is enriched for other “mimetic” cells subsets, such as keratinocyte-like TECs, (neuro)endocrineTECs, myoTEC, etc. these recently described rare subsets should be mapped most of these points could be addressed by using the proposed gating strategies and validating the corresponding gene signatures by bulkRNAseq.

The reviewer asks whether our analysis also identified cell surface markers that recognize several of the recently described “mimetic” mTEC subsets. Unfortunately, none of the transcriptomic features characterizing individual “mimetic” TEC subsets represent sequences that encode cell surface proteins. Since the antibodies used in the LEGENDScreen panel are exclusively directed against epitopes on the cell surface we have not been able to identify markers that correspond to individual “mimetic” mTEC subsets. Nonetheless, we analysed the expression of informative gene transcripts characterising tuft, microfold, enterocyte/hepatocyte, neuroendocrine, ciliated, ionocyte, keratinocyte, and muscle “mimetic” mTEC within our CITEseq dataset. As expected, the detection of transcripts identifying tuft cells correlated with the cluster assignment of tuft-like mTEC. Transcripts characterizing microfold-, enterocyte/hepatocyte-, and keratinocyte-like TEC were detected in clusters G/7-8 within the mTEC^{high} population. Due to the low cell number positive for those genes unsupervised clustering did not assign specific subclusters. This information is now included in the revised manuscript (see Lines 348-354 and new Figure S7).

REVIEWERS' COMMENTS

Reviewer #1 (expert in mTECs, AIRE, transcriptional regulation of T cell development):

The authors have addressed all my concerns in clarifying key points by performing the required additional analyses and discussing the limitations of some of them. They improved visual elements of some figures in following my suggestions and added helpful new fig/tables. Together, the modifications/additions provided by the authors have significantly improved the manuscript in its new version.

Reviewer #2 (expert in massively parallel flow cytometry):

All requested changes have been made to my satisfaction. Thanks for including the extra details in methodology and supplementary tables.

Reviewer #3 (expert in mTECs, transcriptomic analysis of mTECs):

the authors have addressed most of the comments and improved the clarity and quality of their manuscript